

# Ensemble flood forecasting considering dominant runoff processes: II. Benchmark against a state-of-the-art model-chain (Verzasca, Switzerland)

Christoph Horat[1,3], Manuel Antonetti[1,2], Katharina Liechti[1], Pirmin Kaufmann[4], and
Massimiliano Zappa[1]

[1]Swiss Federal Institute for Forest, Snow and Landscape Research, Birmensdorf, Switzerland
[2]University of Zurich, Department of Geography, Zurich, Switzerland
[3]ETH, Institute for Atmospheric and Climate Science, Zurich, Switzerland
[4]MeteoSwiss, Swiss Federal Office for Meteorology and Climatology, Locarno, Switzerland

*Correspondence to:* Massimiliano Zappa (massimiliano.zappa@wsl.ch)

**Abstract.** Model benchmarking is needed in order to establish how newly developed forecasting approaches perform against current state-of-the-art systems. In many cases, resources for re-forecasting long periods of time are limited and therefore, a period of parallel-operations is evaluated. For this study, the forecasting chain presented in the companion paper by Antonetti et al. (2018) has been set up for the Verzasca basins in the southern Swiss Alps. In this region, an operationally running sys-

tem is available from previous studies on probabilistic flash flood (FF) forecasts. This current system relies on the calibrated semi-distributed hydrological model PREVAH. The new model RGM-PRO includes the concept of dominant runoff processes and requires a priori estimation of parameters but no direct discharge observations for calibration. This is a significant benefit to FF prediction in ungauged catchments.

Both FF forecasting chains are forced by information from numerical weather prediction COSMO-1 and COSMO-E. Real-

time rainfall is provided by the CombiPrecip product, which combines rain gauge and weather radar data. As RGM-PRO is an event-based model, initial conditions are not computed internally. Such initial conditions are obtained from operationally available gridded simulations of the PREVAH model. The current PREVAH-HRU setup uses rainfall data as obtained by interpolating real-time data of the station network maintained by MeteoSwiss. Initial conditions are tracked internally day-by-day.

The PREVAH-HRU runs forced by COSMO-1 and COSMO-E during real-time operations in the period May to August

2016 have been compiled. Corresponding model runs using RGM-PRO have been computed a posteriori. Both sets of forecasts are evaluated against discharge observations using deterministic and probabilistic verification metrics.

Results showed that the novel approach was able to compete with the operational benchmark prediction system and was consistently superior for high-flow situations. The new forecasting chains were able to react faster on precipitation in comparison with the benchmark forecasts. Confirming previous studies for all forecasting chains, a clear preference for using a

meteorological ensemble as forcing data was found. In a synthesis of the two companion papers, more skill was found in the Verzasca basin than in the Emme catchment, suggesting a better forecast performance in strongly topography driven basins with shallow soils and weak dependence on initial conditions.



The findings of the two studies suggest that the novel forecasting chains can compete with the traditional ones in operational setup without the need of long-term discharge measurements and extensive calibration. With the new runoff generation module, extension of FF prediction to ungauged catchments is possible, provided that spatially distributed information on dominant runoff processes is available.

# 1  Introduction

During or minutes to hours after heavy precipitation strikes a catchment, a pronounced maximum in stream flow may evolve in the riverbed (Norbiato et al., 2008). These so-called flash floods (FFs) are of high societal relevance, especially in alpine areas where topography promotes heavy precipitation (Panziera and Germann, 2010). For this reason, there is a strong need for skilful operational predictions, to enable early warnings and to increase the flood damage mitigation capabilities (Rossa et al., 2011). Whilst the companion paper (Antonetti et al., 2018) discusses current approaches and challenges in FF forecasting from a hydrological perspective, this second study focusses more on the meteorological perspective and the benchmark against an established operational hydrometeorological prediction system. As FFs always arise from an interplay of the hydrosphere and atmosphere, it is important to keep the perception of a coupled system in mind.

The occurence of FFs is closely linked to heavy precipitation (Norbiato et al., 2008). Therefore, one method for predicting FFs is to predict heavy precipitation events (HPEs). As there is usually only little time to initiate a FF alert (Liechti et al., 2013a), computationally efficient methods are preferred for operational prediction systems. In an early work, Doswell et al. (1996) pursued an approach for FF forecasting purely based on meteorological conditions. The basic conditions for a HPE to occur are identified, namely ascent of moist air and subsequent condensation. A HPE is characterised by an extraordinarily large amount of total precipitation at a certain location, which is the product of rainfall rate and rainfall duration. In their simplified approach, they assume rainfall rate to be proportional to the magnitude of the vertical moisture flux. The duration depends on the movement of the system, e.g. it is large when a convective cell remains stationary at a certain place. The problem with FFs is that they occur too infrequently that a forecaster could develop enough experience with them. Therefore, the approach of Doswell et al. (1996) is thought to be a support for forecasters. They can anticipate the possibility of a FF when specific meteorological conditions are met, whereby it is assumed that every FF shares some common features concerning the generation mechanism. However, the approach of Doswell et al. (1996) does not include rainfall-runoff processes for FF prediction. The authors acknowledge that the development of a FF depends on factors such as antecedent precipitation, size and topography of the drainage basin, land use etc.

A more quantitative approach that is still based mainly on meteorological parameters is the one of Alfieri et al. (2011) that uses numerical weather prediction (NWP), radar-NWP blending and radar nowcasting. First, Alfieri et al. (2011) derived an index for the severity of accumulated upstream forecast precipitation based on 30 years of hindcast climatology simulated with the atmospheric model COSMO (COnsortium for Small-scale MOdeling (Marsigli et al., 2005; Montani et al., 2011)). This index implicitly takes river network into account and can be used to detect areas where HPEs with a large return period are forecast. In a more advanced stage of the event, more accurate information is provided as radar-NWP blended and radar now-



casting products are used to issue warnings. Although their method does not rely on calibration and could therefore be applied to ungauged catchments, it is only suitable for catchments with areas up to 1000-2000 km$^2$ and does not give any information on timing and magnitude of the event.

A similar method was developed by Panziera et al. (2016), carrying out an extreme rainfall analysis based on 10 years of radar data in Switzerland and derived thresholds for an automatic alert system. If the sum of past and predicted precipitation exceeds the defined threshold values, a warning for the catchments ranging from 100 to 500 km$^2$ in area is issued.

Although the above-mentioned methods are alluring due to their simplicity and therefore potentially strong utility for operational applications, heavy rainfall is a necessary – yet insufficient – criterion for the occurence of FFs, particularly for small scale catchments (Norbiato et al., 2008). The hydrological state of the system, in particular the antecedent soil moisture, as well as the infiltration capacity of soils and interception, play a key role. Therefore, hydrological and meteorological models are coupled in forecasting chains and complemented with nowcasting tools for initial conditions and warnings for end-users. Relevant examples of forecasting chains are described below, with a particular focus on the meteorological aspects.

Zappa et al. (2008) presented several end-to-end forecasting system for alpine flood events as developed within the Mesoscale Alpine Programme Demonstration of Probabilistic Hydrological and Atmospheric Simulation of Flood Events (MAP D-PHASE). In total, over 30 different hydrological and meteorological models are combined in over 60 catchments, with a particular focus on probabilistic forecasts. A first example that is mentioned here is the combination of COSMO-2 with FEWS/HBV (Flood Early Warning System/Hydrologiska Byrns Vattenbalansavdelningels), which was the hydrological model of Federal Office for the Environment (FOEN) at that time. With high-frequent updating and temporal overlapping of deterministic COSMO-2 runs a so-called time-lagged ensemble is generated and used for a pseudo-probabilistic forecast. As a second example, DIMOSOP (DIstributed hydrological MOdel for the Special Observing Period) is combined with COSMO-LEPS for a small flood in the Oglio basin in the Central Italian Alps. Furthermore, Zappa et al. (2008) were possibly the first that coupled a hydrological model with a real-time radar ensemble. The radar ensemble is generated by combining stochastic simulation and knowledge of radar error covariance structure (Germann et al., 2009) and helps to assess sensitivity to and uncertainty of initial conditions in hydrological models. In addition, large efforts were made to enhance communication between scientists, warning agencies and task forces responsible for flood management, e.g. with workshops for end-users at different stages of the project and a visualisation platform (Zappa et al., 2008; Rotach et al., 2012). Alert thresholds were then determined from discussions between scientists and end-users. For atmospheric models, alerts are based on 3, 6, 12, 24, 48, and 72 hours accumulated precipitation. For hydrological models, alerts are based on hourly river runoff predictions.

In a case study for the 26 September Venice FF, Rossa et al. (2010) implemented a forecasting chain using the COSMO-2 meteorological model. The innovation in their approach is the assimilation of radar quantitative precipitation estimate (QPE) into COSMO-2 via the latent heat nudging method, resulting in improved initial conditions for the meteorological model. With this method, intense convection is triggered at the correct location and incorrect precipitation is suppressed. It is therefore ensured that the main convective systems are introduced in the model, which has a positive impact on forecast quality for about 2-5 hours (Rossa et al., 2010).



Precipitation predictions relying only on real-time radar data are significantly disadvantageous when considering the short response time of small catchments prone to FFs (Alfieri et al., 2011; Liechti et al., 2013a). To extend lead time, heuristic methods exist to issue forecasts with radars, namely Eulerian and Lagrangian persistence. Eulerian persistence takes the current radar image as a forecast for the near future (Germann and Zawadzki, 2002), whereas Lagrangian persistence extrapolates the

current radar image with the past motion of the precipitation (Germann and Zawadzki, 2004). Liechti et al. (2013a) set up two radar-based ensemble forecasting chains for FF prediction in alpine catchments in southern Switzerland including the Verzasca basin, which is subject of this study. The first ensemble forecasting chain uses NORA (Nowcasting of Orographic Rainfall by means of Analogues, Panziera et al. (2011)) as meteorological input, where the current situation is compared with analogues of an archive. Subsequently, the 12 members of the archive with most similar mesoscale flow, air-mass stability and radar

fields are issued as an ensemble forecast for the next eight hours. The second ensemble forecasting chain uses a connection of REAL (Radar Ensemble generator designed for the Alps using LU decomposition) and COSMO-2 as meteorological input (REAL-C2). REAL is a nowcasting tool which generates a radar ensemble of 25 members by adding stochastic perturbations to the current radar field. For the perturbation field, detailed knowledge about space-time variance and auto-covariance of radar errors must be known. This is combined with stochastic simulation techniques (Germann et al., 2009). REAL initially forces

the hydrological model whereas at a later time step COSMO-2 is used as meteorological input. For verification, Liechti et al. (2013a) investigated, besides the two radar-based ensemble forecasts mentioned, three additional deterministic forecasting chains. They find a clear superiority of the two chains using ensembles, with REAL-C2 being the forecasting chain that performs best. However, NORA cannot be computed efficiently in real time, which is the reason why this approach is not suitable for operational applications.

A forecasting chain which is of high practical relevance for operational flood early warning for the city of Zurich was implemented by Addor et al. (2011) for the river Sihl. They couple the PREVAH hydrological model with a hydraulic model and use deterministic COSMO-7 and probabilistic COSMO-LEPS as meteorological input. An analysis of hindcasts is performed for the period of June 2007 to December 2009. In a similar way as Liechti et al. (2013a), they find a clear preference for the probabilistic forecast for all lead times and event intensities investigated. However, Addor et al. (2011) find skill of their

approach to be limited for medium lead times and high threshold quantiles. In particular, forecast performance is relying on accurate precipitation predictions, as the Sihl catchment is relatively small ($336\ \text{km}^2$).

In order to benefit from ensemble rainfall predictions, Rebora et al. (2006) performed stochastic downscaling as input for a semi-distributed rainfall-runoff model in regions where a full small-scale deterministic model was not available. Their forecasting chain relies on Lokal Model (previous name of COSMO) as meteorological input. Another innovative approach was

introduced by Kim and Barros (2001), who trained a neural network on relationships among convective weather systems, rainfall production and streamflow response and produced a skilful forecast of FFs for Pennsylvania.

## 1.1 Challenges and uncertainties

Due to the strong non-linearity of the investigated system, the prediction of FFs remains challenging (Rossa et al., 2011). To represent convective systems that are responsible for heavy rainfall, a high spatial and temporal model resolution is needed




which requires substantial computer power (Collier, 2007). With complex topography, the setup is further complicated (Zappa et al., 2011). Considering all model components, *meteorological uncertainty* is usually the dominant source contributing to total uncertainty (Rossa et al., 2011; Zappa et al., 2011). For larger catchments with long response time it may be sufficient to use precipitation nowcasts as still enough time to issue warnings remains (Liechti et al., 2013a). A prediction system then

can benefit from the advantages of radar, which is a commonly used nowcasting tool (Rossa et al., 2010). In contrast, for smaller catchments, precipitation forecasts are crucial (Liechti et al., 2013a). However, measuring the spatial and temporal distribution of the current precipitation is demanding, and knowing the space-time field of rainfall in advance is even more challenging. Convective systems are often not represented in a satisfying manner in NWP (Liechti et al., 2013a). As quality of meteorological input is essential for forecasting chains, Ehret (2011) compared catchment-averaged rainfall forecasts with

ground-level observations in seven mesoscale alpine catchments in Bavaria for five operational models: GME (Globalmodell Europa), COSMO-EU (COSMO Europa), GFS (Global Forecast System), ALADIN-Austria and COSMO-LEPS. He found a clear preference for the median of COSMO-LEPS when comparing with deterministic forecasts and the COSMO-LEPS ensemble performed better than a poor man's ensemble built from GME, GFS and COSMO-EU.

Additional difficulties arise for operational FF prediction, as computational resources are limited (Rossa et al., 2011) and the

system should be easily transposable to various catchments.

## 1.2  Objectives

Forecasts based on meteorological parameters alone are alluring as they are promising broad applicability in operational use. However, findings of Doswell et al. (1996), Alfieri et al. (2011) and Panziera et al. (2016) showed that FF prediction based on meteorological parameters alone is not enough. Quantitative FF forecasts are needed and can be provided by coupling

meteorological and hydrological models. In order to expand FF prediction to ungauged catchments, Antonetti et al. (2017) introduced RGM-PRO, a new hydrological module with process-based runoff generation and no need for calibration (Antonetti et al., 2018). In this study, FF forecasting chains are set up using high-end model components, namely the RGM-PRO runoff generation module, atmospheric models COSMO-1 and COSMO-E and CombiPrecip nowcasting product for the Verzasca region in southern Switzerland. As there is no prior experience with using RGM-PRO for FF early warning, the skill of the

new forecasting chains is evaluated for summer 2016 and compared with an already operationally running system (e.g. Zappa et al. (2011, 2013)) that serves as a benchmark. Thus, the main research question of this study:

> Is it possible to increase skill and extend operational use in FF prediction with a forecasting chain that includes a newly developed conceptual hydrological module with process-based runoff generation and no need for calibration (RGM-PRO) compared with existing forecasting tools?

The target area and the ensemble prediction chains are further described in Sect. 2 and 3, respectively. In Sect. 4, the data analysis and verification methods can be found. The results are presented in Sect. 5 and discussed in Sect. 6. In Sect. 7, the conclusions are drawn.



## 2    Target area

The Verzasca catchment, depicted in Fig. 1, lies in the southern Alps in the Canton of Ticino, Switzerland. It ranges from 490 to 2900 m a.s.l. and covers an area of 186 km$^2$ with relatively little anthropogenic influence (Liechti et al., 2013a). The Pincascia subcatchment (44 km$^2$) is located in the Eastern part of the main catchment. In terms of land use, 30 % of the Verzasca catchment consists of forest, 25 % of shrub, 20 % of rocks and 20 % of alpine pastures (Wöhling et al., 2006). Considering geological properties, the bedrocks are mostly crystalline gneisses with some calcareous shists (Bündnerschiefer) and the lithology of the Pennine units of the central Alps is dominant in the basin (Georg et al., 2006).

## 3    Ensemble prediction chains

### 3.1    Hydrological models

For the operational benchmark prediction system run by WSL, the traditional PREVAH hydrological model (Viviroli et al., 2009) was used. PREVAH stands for Precipitation-Runoff-EVApotranspiration HRU model, where HRU stands for Hydrological Response Unit. On the left hand side of Fig. 1, HRUs for the Verzasca catchment are indicated. The traditional version of PREVAH relies on long-term discharge measurements for calibration.

In order extend FF prediction to ungauged basins, i.e. having a hydrological model that is not depending on runoff data, Antonetti et al. (2017) developed RGM-PRO, which is an advancement of the runoff generation module of the traditional PREVAH. RGM-PRO stands for Runoff Generation Module PROcess-based and includes spatially distributed knowledge on dominant runoff processes (DRPs) in maps of runoff types (RTs), which allows to determine the model parameters a priori. The attribution of DRPs and RTs is detailed in the companion paper Antonetti et al. (2018) and in Antonetti and Zappa (2017). Various approaches with different amount of complexity exist to generate the maps of RTs (Antonetti et al., 2016a). For our study area, an automatic and therefore relatively simple method described by Müller et al. (2009) was used, with the corresponding map of RTs - also referred to as Müller map - shown in Fig. 1 on the right hand side.

As visible in Fig. 1, in the traditional operational implementation of PREVAH the 500 m grid points are first aggregated to HRU (according to Gurtz et al. (1999) and Viviroli et al. (2009)). This reduces the nominal resolution of the simulations and thus generalises the local runoff generation behaviour. In RGM-PRO, a sub-grid parametrisation is introduced to better account for the local differences in DRPs. This increases the nominal resolution of the application. Furthermore, the meteorological input for the original PREVAH application is generalised according to sub-areas and elevation bands (e.g. Liechti et al. (2013b)). RGM-PRO requires hourly gridded precipitation input and runs at a spatial resolution of 500 m. A more detailed description of the hydrological models mentioned here can be found in Viviroli et al. (2009), in the companion paper Antonetti et al. (2018) or in Antonetti et al. (2017).



## 3.2 Meteorological and hydrological data

### 3.2.1 Observed data

Following the setup presented in the companion paper (Antonetti et al., 2018), CombiPrecip (Sideris et al., 2014) has been used as high resolution hourly gridded precipitation product. CombiPrecip was used here only for forcing RGM-PRO. The operational prediction chain using PREVAH is embedded in the real-time data flow adopted since MAP D-PHASE (Zappa et al., 2008). Hourly data from different surface monitoring networks are collected from various providers including MeteoSwiss, the Swiss Federal Office for Climatology and Meteorology, and several regional administrations. The real-time information of observed climate variables (e.g. rainfall, air temperature) is updated each hour and stored into a database of the WSL Institute for Snow and Avalanche Research SLF. The SLF database also includes data from a dense network of stations located at high elevations (Romang et al., 2011). The data needed to force the operational PREVAH chain are then interpolated according to the procedures described in Liechti et al. (2013a), Andres et al. (2016) and further previous studies with the PREVAH model. The adopted techniques for precipitation interpolation is the inverse distance weighting (Viviroli et al., 2009). The hourly runoff measurements needed to evaluate the experiments were provided by FOEN.

### 3.2.2 Numerical weather predictions

MeteoSwiss developed a configuration of the COSMO model (Marsigli et al., 2005; Montani et al., 2011) with 1.1 km grid spacing, the COSMO-1. It runs as deterministic model and is initialised from its own assimilation cycle using the nudging scheme. Forecasts are calculated every three hours in a rapid update cycle with a forecast range of 33 hours and once per day (03 UTC forecast) out to 45 hours. This setting was operationalised in spring 2016 and replaced the former COSMO-2 with 2.2 km grid spacing. As its predecessor, COSMO-1 assimilates radar-derived QPE using latent head nudging. Latent heat nudging is able to considerably increase the accuracy of the precipitation forecast during the first 6 to 12 hours of the forecast. The boundary conditions are taken from the newest available ECMWF (European Centre for Medium-Range Weather Forecasts) high resolution forecast (HRES).

In addition to the deterministic COSMO-1, the ensemble system COSMO-E with 2.2 km grid spacing was operationalised in May 2016. It is initialised twice per day and has a lead time of 120 hours. The assimilation cycle uses an ensemble transform Kalman filter approach. The boundary conditions are taken from randomly selected 20 members of the ECMWF ensemble forecast (ENS). It uses the SPPT scheme to simulate the effect of the model uncertainty. At MeteoSwiss, COSMO-E replaces COSMO-LEPS that has a lower resolution with 7 km grid spacing.

## 3.3 Activation of RGM-PRO

As RGM-PRO is an event-based model, simulations were not computed on each day of the investigation period but only on specific days, so-called *alert dates*. In order to assess susceptibility of a basin to FFs, a method was developed with a combination of soil moisture data from PREVAH simulations and precipitation forecasts from COSMO-LEPS. Please note





that COSMO-LEPS and not COSMO-E was used, as a longer time series of data was available. Several combinations with threshold exceedances of maximum daily accumulated precipitation of a 500x500 m grid cell of one, two or three ensemble members of COSMO-LEPS were investigated. Furthermore, the methods were analysed without taking soil moisture data into account. A day was considered an alert date when soil moisture one day before and precipitation forecast for the next day

were higher than certain thresholds. These thresholds were varied in order to minimise false alarm ratio (FAR) and still have probability of detection (POD) equal one (see Sect. 4) for May to August 2010-2016, where an hourly specific runoff larger than 1 $\frac{m^3}{km^2 \cdot s}$ was defined as event. This resulted in a total of 22 alerts from May to August 2016. This procedure ensured that the performance of the model was not only evaluated for events but also during non-events, for which it is important to see whether a forecasting chain generates a false alarm.

### 3.4  Overview of completed experiments

This study followed the experimental setup depicted in Fig. 2. RGM-PRO was set up with Müller map and combined with COSMO-1 and COSMO-E, building the forecasting chains DRP-C1 and DRP-CE. During the initialisation period, gridded precipitation fields from CombiPrecip and soil moisture data from PREVAH were used. Onset of initialisation took place at the moment in time with minimum observed runoff in the last five days prior to the forecast. In order to assess the quality of

the meteorological forecast, RGM-PRO reference runs for DRP-C1 and DRP-CE were exclusively forced with CombiPrecip data (see supplementary material).

For our study area, an operational hydrological forecasting system run by WSL already exists (e.g. Zappa et al. (2011, 2013)). The traditional, calibrated PREVAH is initialised with six meteorological variables and combined with COSMO-1 and COSMO-E, onwards referred to as TRAD-C1 and TRAD-CE. The traditional forecast were always initialised exactly five days

prior to the switch to forecast mode according to the operational initial states of the real-time system. This established chain uses precipitation input interpolated by inverse distance weighting from operationally available pluviometers (Andres et al., 2016). Comparing the newly developed with the traditional forecasting chains will show, on the one hand, possible benefits of including knowledge on DRPs into hydrological modelling. On the other hand, differences may arise due to the use of CombiPrecip instead of pluviometer data for model initialisation. A comparison of forecasting chains fed with COSMO-1 and

COSMO-E will show whether high resolution deterministic or lower resolution probabilistic NWP data is favourable.

Our investigation period was confined to May until August 2016, as COSMO-E and COSMO-1 are available only since March 2016. Furthermore, events earlier than May are usually affected by meltwater and therefore could not be simulated with RGM-PRO. Please note that the availability of runoff data was different for the Verzasca and the Emme region (the latter is investigated in the companion paper (Antonetti et al., 2018)). At each alert date, which are explained in Sect. 3.3,

deterministic forecasting chains were run eight times with each COSMO-1 forecast available on that day and probabilistic chains two times with each COSMO-E forecast. This resulted in a total of 5'280 hours of forecast for each forecasting chain based on COSMO-1 and yielded 5'016 forecast-observation pairs for each chain that was relying on COSMO-E. However, when comparing deterministic and probabilistic systems for a specific lead time there was four times less data for the ensemble approach.





The strength of the approach presented here is the high operational utility, as no calibration of the hydrological model and therefore no runoff data is needed. Furthermore, the system is easily transposable and can be applied in any region where the necessary data is present to generate a DRP map (Antonetti et al., 2016a), provided that appropriate meteorological input and soil moisture data is available as well. As precipitation forecasts and not only nowcasts are included, lead time to issue warnings

is extended. Meteorological uncertainty is treated with the ensemble approach to account for spread in timing, location and intensity of rainfall (Addor et al., 2011; Liechti et al., 2013a).

## 4   Data analysis

Forecast verification in general is based on methods described by Wilks (2011) and in particular follows the suggestions by Brown et al. (2010) on applications for hydrological ensemble predictions. The strategy adopted here closely follows the ideas

proposed in Addor et al. (2011) and Liechti et al. (2013a).

### 4.1   Deterministic continuous forecasts

Deterministic continuous forecasts were turned into deterministic forecasts for dichotomous predictands, where event and non-event were distinguished with a threshold which was a quantile of hourly runoff climatology from May to August 2016. In this case, both forecast and observation could have values of 0 or 1. For deterministic forecasts for discrete predictands, the Brier

score (BS), the Brier skill score (BSS), the POD, the FAR, the probability of false detection (POFD) and the frequency bias (FB) were calculated. The BS measures the correspondence of threshold exceedance for forecast and observation but does not take into account magnitude of difference. A perfect prediction delivers a value of zero for BS and a BSS of one. For BSS (Eq. 1), the mean runoff from May to August 2016 served as reference forecast (Eq. 2).

$$BSS = 1 - \frac{BS}{BS_{clim}} \qquad (1)$$

with

$$BS_{clim} = \frac{1}{n} \sum_{k=1}^{n} (\bar{o} - o_k)^2 \qquad (2)$$

The POD is the number of times a threshold exceedance was correctly forecast ("hit") divided by the number of times a threshold exceedance occured. The FAR is the number of cases where a threshold exceedance was forecast but did not occur ("false alarm"), divided by the total number of forecast threshold exceedances. Similarly, the POFD is the number of false

alarms divided by the number of observed non-events. A perfect forecast has a POD of one, while the FAR and the POFD equal zero. The FB is the number of forecast threshold exceedances divided by the number of observed threshold exceedances. It does not indicate how well forecast threshold exceedances and observed threshold exceedances correspond in time but whether threshold exceedances in general are over- or under-forecast.





Continuous deterministic forecasts investigated in this study were simulations driven by COSMO-1 and the median in hydrographs obtained using COSMO-E as forcing.

## 4.2 Probabilistic continuous forecasts

Probabilistic continuous forecasts, as they resulted from simulations with COSMO-E, were turned into probabilistic forecasts

for discrete predictands, where again a quantile of climatology served as threshold to distinguish between event and non-event. If, for instance, 17 out of 21 ensemble members were higher than the threshold, the probability of threshold exceedance was $p = \frac{17}{21}$, as all ensemble members are equally likely. Probabilities were rounded to one decimal place. For probabilistic forecasts for discrete predictands, BS and BSS were computed, as the BSS allows a direct comparison between deterministic and probabilistic forecasts. Furthermore, BS and BSS were decomposed into reliability, resolution and uncertainty contributions.

*Reliability* measures the relationship of the forecast to the distribution of observations for one specific forecast value. A reliable (i.e. good) forecast has a small reliability term. *Resolution* measures how well distributions of observations for different forecasts can be distinguished. Ideally, resolution is high. The third contribution is *uncertainty*, which can not be influenced by the forecaster but is completely determined by the nature of the event. Reliability, resolution and uncertainty were visualised as calibration function and refinement distributions in reliability diagrams that are found in the supplementary materials.

In addition, probabilistic forecasts for dichotomous predictands were turned into deterministic forecasts for dichotomous events with varying probability thresholds. A probability below the threshold was turned into a 0 % likeliness and a probability above the threshold was turned into a 100 % likeliness for event-occurrence. For various probability thresholds, POD and POFD were calculated and visualised as a curve in a receiver operating characteristics (ROC) diagram. The area under a ROC curve (ROCa) is a measure of discrimination and is 1 for a perfect forecast and 0.5 for a useless forecast. The minimum value

of ROCa useful for end-users is 0.7 according to Buizza et al. (1999). For different threshold quantiles and lead times, values of ROCa are compiled in a summary (see Fig. 8), which allows a comparison of approaches and catchments.

## 4.3 Bootstrapping

To assess the sampling uncertainty of skill score computations the bootstrapping approach described by Efron (1979) was used with 500 iterations, which enabled visualisation of skill scores as boxplots. As time windows of 6 to 24 hours were considered,

assumption of independence may not be strictly valid in our case and a moving-window bootstrap could be more appropriate. However, this method was not implemented to ensure comparability with Liechti et al. (2013a).

## 4.4 Peak-box approach

The peak-box approach was introduced by Zappa et al. (2013) and can be used to estimate timing and magnitude of runoff peak for probabilistic forecasts. For every member of the ensemble in hydrographs, magnitude and timing of the respective peak flow

was computed, which lead to 21 peaks for COSMO-E. A so-called peak-box was then drawn around the 21 ensemble peaks confined to the left by the earliest occuring ($t_0$), to the right by the latest predicted ($t_{100}$), to the bottom by the lowest occuring





($p_0$) and to the top by the highest predicted ($p_{100}$) peak. In addition, a second box, referred to as IQR-box, was depicted with the 25 %- and 75 %-quantiles in terms of peak timing ($t_{25}$ and $t_{75}$) as confining x-coordinates and the 25 %- and 75 %-quantiles in terms of peak magnitude ($p_{25}$ and $p_{75}$) as confining y-coordinates. The best estimate for the peak was then chosen as the point $P_{BE}$ ($t_{50}$, $p_{50}$) with the 50 %-quantile in terms of peak timing ($t_{50}$) as x-coordinate and with the 50 %-quantile in terms of peak magnitude ($p_{50}$) as y-coordinate.

As a skill metric, absolute difference of timing ($D_{TIME}$) and magnitude ($D_{PEAK}$) between $P_{BE}$ and observed peak, which is referred to as $P_{OBS}$ ($t_{obs}$, $p_{obs}$), were computed. In addition, it was calculated how often observed peak was within the IQR-box ("IQR hit"), how often inside the peak-box (but not within IQR-box, referred to as "peak-box hit") and how often outside the peak-box ("no hit"). Furthermore, the areas of the peak-box and the IQR-box serve as a measure of uncertainty of the forecast, which can therefore be calculated with

$$UC_{peak-box} = (p_{100} - p_0) \cdot (t_{100} - t_0) \cdot \frac{3.6}{A} \tag{3}$$

and

$$UC_{IQR-box} = (p_{75} - p_{25}) \cdot (t_{75} - t_{25}) \cdot \frac{3.6}{A} \tag{4}$$

Units of $UC_{peak-box}$ and $UC_{IQR-box}$ are millimetres and $A$ in Eq. 3 and 4 stands for the catchment area in km$^2$. Evaluation with the peak-box approach was done for the two different probabilistic forecasting chains (DRP-CE and TRAD-CE) with 44 cases each in the Verzasca and Pincascia catchments.

## 5 Results

### 5.1 General assessment of the used numerical forecasts

The NWP model COSMO-2 and its successor COSMO-1 were both operational during the summer season (JJA, also denoted as s3) in 2016. This allows for a one-to-one comparison of the two models. The POD and FAR of the predicted 12 hour precipitation sums of both models are shown in the upper panel of Fig. 3. These values are determined using the automated rain gauge network of Switzerland. While the two models are nearly indistinguishable for the 1 mm/12h threshold, COSMO-1 has a lower FAR for most of the higher thresholds. The POD however is also reduced, especially for the highest threshold (50 mm/12h). The reason for this behaviour can be seen in the lower panel showing the FB, which is lower in COSMO-1 for the larger thresholds and especially for the largest. The FB of COSMO-1 is closer to 1 and indicates a better performance than COSMO-2, but the almost inevitable effect of a reduced FB is to also reduce both POD and FAR of the forecast. Note that the values get more uncertain to the right of Fig. 3 (not indicated), where the events get increasingly rare. Also note that COSMO-1, with its twofold smaller grid spacing, is more affected by the double penalty effect (double-counting of error for misplaced rain cells) in a rain-gauge-based, pointwise verification than COSMO-2, a fact that might hide part of the true performance benefit of COSMO-1.





The ensemble prediction systems COSMO-LEPS and COSMO-E are compared in Fig. 4 in terms of BSS for two precipitation thresholds. When looking at the same season for both models, indicated by the same color, it is always COSMO-E (triangle) that is above COSMO-LEPS (dot), indicating that COSMO-E is always better than COSMO-LEPS in terms of BSS. As the stronger structured convective precipitation in summer is more difficult to predict than the more often synoptically driven

precipitation in spring (MAM, also denoted as s2), the respective BSS values for summer are always lower than those for spring. They are however only available for COSMO-E and cannot be compared to COSMO-LEPS, for which operational service was ceased.

As temperature is another important factor of hydrological forecasting, the two ensemble models are compared in Fig. 5 in terms of one relevant temperature threshold. As with precipitation, COSMO-E almost always has a better score than COSMO-

LEPS and only for the very short lead time, the two are virtually equal. For temperature, the variability in spring is higher than in summer and thus more difficult to predict, as can be seen in Fig. 5 where the summer values, only available for COSMO-E, are always clearly above the respective spring values.

## 5.2 Ability to detect events and reject non-events

For the Verzasca catchment, POD, FAR and FB are depicted in Fig. 6 for various threshold quantiles and lead times. There is

not much difference between DRP-C1 and DRP-CE (med) and between TRAD-C1 and TRAD-CE (med). The main difference is the one between process-based forecasting chains and traditional forecasts. Furthermore, there is not much change of the pattern with lead time, except that FBs for high quantiles of process-based forecasts are growing over time.

For all threshold quantiles, POD but also FAR of process-based forecasting chains are higher than the ones of the traditional forecasts. POD for process-based forecasts remain close to one for high quantiles whereas traditional forecasting chains only

detect around every second event. In turn, FAR for traditional forecasts are close to the ideal value of zero, whereas FAR for process-based forecasts have a peak around the $q_{0.7}$ quantile but get low for the very high threshold quantiles. This leads in summary to a better performance of the process-based forecasts for the very high quantiles as POD are much higher but FAR only slightly higher.

In terms of bias, traditional forecasts reveal strong under-forecasting of events especially for higher quantiles. Process-

based forecasting chains exhibit almost perfect bias for low to medium quantiles but substantial over-forecasting of high-threshold events. Results for the nested Pincascia catchment can be found in the supplementary materials. They reveal similar characteristics as in the Verzasca basin although FAR in Pincascia is higher especially for low quantiles.

## 5.2.1 Comparison of deterministic and probabilistic forecasts

Investigation of BSS in Verzasca catchment indicates that there is skill for DRP-C1, TRAD-C1, DRP-CE and TRAD-CE for

all quantiles up to a lead time of 29 hours (Fig. 7). A tendency of decrease in BSS with increasing lead time is not clearly visible for the lower quantiles but for the higher ones. Furthermore, uncertainty bands of BSS resulting from the bootstrapping approach are usually more or less constant with lead time except for the $q_{0.99}$ quantile, where spread of DRP-CE and TRAD-CE decreases with lead time. For higher quantiles, uncertainty in BSS is usually larger and spread is substantial for the highest



threshold quantiles.

Whether DRP-C1 performs better than TRAD-C1 or vice versa is highly sensitive on the chosen threshold quantile and cannot be safely determined. In contrast, DRP-CE has higher values of BSS than TRAD-CE in most cases, in particular for the high threshold quantiles. Exceptions are found for the $q_{0.9}$ threshold quantile, where DRP-CE and TRAD-CE perform

comparable, and for the longest lead times of the $q_{0.7}$ threshold quantile, where TRAD-CE is better than DRP-CE. In general, for short lead times and the highest quantiles process-based forecasting chains have higher values in BSS than traditional forecasts. This "starting gap" is significant for the $q_{0.8}$ and $q_{0.95}$ threshold quantiles.

Comparing deterministic and probabilistic forecasting chains shows superior performance of DRP-CE over DRP-C1 in most cases, where DRP-C1 is most competitive at very short lead times. These characteristics hold as well for the traditional forecast.

When comparing BSS for deterministic and probabilistic forecasting chains for lead times up to 29 hours, a very similar picture results in the Pincascia basin (see supplementary materials).

### 5.3    Synthesis of the forecast quality

In the Verzasca (Pincascia) basin, forecasts are of use for decision makers (ROCa value larger than 0.7, (Buizza et al., 1999)) up to 96 (72) hours when considering the $q_{0.99}$ threshold quantile (Fig. 8). The summaries of ROCa depict a clear preference

for DRP-CE over TRAD-CE for the highest quantiles in both catchments. Only exception is the $q_{0.99}$ threshold quantile for 48 hours lead time in the Pincascia basin. In contrast, TRAD-CE is favoured for the two lowest quantiles. Preference of DRP-CE over TRAD-CE is more strongly pronounced in Verzasca catchment compared with Pincascia. Furthermore, forecasts are useful for more threshold quantiles and lead times in Verzasca basin when compared with Pincascia catchment. However, predictions of longest lead times and highest threshold quantiles are not of use in both catchments.

For both catchments, the values of ROCa depict a strong drop in quality for increasing lead times (see supplementary materials).

### 5.4    Analysis of a particular event

The deterministic discharge prediction for the largest event observed in the Verzasca basin during the study period (June 13[th]-17[th] 2016) is shown in Fig. 9. In general, when comparing with TRAD-C1, DRP-C1 reaction on rainfall is more intense generating higher values in runoff. On the one hand, this is visible for the reference run in the DRP-C1 panel (RGM-PRO

forced with CombiPrecip) when comparing it with the one in the TRAD-C1 visualisation (calibrated PREVAH forced with pluviometer data). On the other hand, this appears in the forecast mode as well, as it leads to an overshoot of some DRP-C1 forecasts for the main peak, while the TRAD-C1 chain predicts more conservatively.

A striking advantage of DRP-C1 over TRAD-C1 in this case is found when investigating the onset of the event: On 15[th] of June, 23:00, when the observed hydrograph starts to rise, most trajectories of the DRP-C1 time lagged ensemble increase as

well. In contrast, it takes several hours until the majority of TRAD-C1 hydrographs starts to rise as well.

In Fig. 10, probabilistic flood forecasts of the same event with DRP-CE and TRAD-CE are depicted. The uppermost panel shows a comparison of CombiPrecip and ensemble rainfall predictions from COSMO-E with a switch to forecast mode at 19:00 on 15[th] of June. Please note that the switch for TRAD-CE is few hours earlier, due to a slightly different operational





setup. The second and third panel depict ensemble area and peak-box plot for DRP-CE, whereas in the fourth and fifth panel the same is shown for TRAD-CE. The main phase of precipitation input is on June 16, with CombiPrecip lying within the predicted ensemble range. The observed peak then occurs at midnight at the end of the precipitation phase, implying that severe runoff can develop quickly in Verzasca basin. Peak runoff is higher than 390 m³/s, which is approximately the 2-year

return period (FOEN, 2018). For DRP-CE, the observed hydrograph lies completely within the ensemble spread, whereas the runoff peak is not captured by TRAD-CE max. The observed peak is captured by the peak-box of both prediction chains, although for TRAD-CE this holds only just. For both forecasting chains, the timing of the best peak estimate is very good but the magnitude is substantially underestimated. Considering the complete re-simulation of the hydrograph with RGM-PRO and CombiPrecip reveals very good performance, with almost perfect agreement for the first few hours on $16^{th}$ of June. In

particular, the performance of the RGM-PRO reference run with CombiPrecip is better than the one of the calibrated PREVAH forced with pluviometer data. As in case of the deterministic forecasts forced with COSMO-1, DRP-CE is found to react quicker and more strongly on rainfall compared with TRAD-CE. Again, this leads to a higher spread for DRP-CE.

## 5.5  Evaluation of peak runoff

Peak timing and peak runoff are two very relevant properties characterising a FF. Some methods for verification of peak

runoff are available (Ehret and Zehe, 2011; Zappa et al., 2013). The peak-box method of Zappa et al. (2013) is so far the only one tailored to ensemble forecasts. The evaluation of the peak-box method - as visualised in Fig. 10 - for all events in the Verzasca and Pincascia basins and both probabilistic forecasting chains is shown in Table 1. In the Verzasca catchment, DRP-CE outperforms TRAD-CE in terms of $D_{TIME}$, IQR hit, peak-box hit and no hit. However, values for the latter three are comparable. The slightly better hit rates for DRP-CE come at the cost of substantially larger uncertainties, revealed by both

higher $UC_{IQR-box}$ and $UC_{peak-box}$ values for the process-based approach. In terms of $D_{PEAK}$, TRAD-CE is to favour over DRP-CE.

In the Pincascia basin, TRAD-CE has slightly lower errors in terms of timing ($D_{TIME}$) and peak magnitude ($D_{PEAK}$) in comparison with DRP-CE. For TRAD-CE, IQR hit rate is substantially higher and no hit rate substantially lower than respective values for DRP-CE. This is the case although uncertainty of IQR-box is more than a factor three higher for DRP-CE. Peak-box

hit rate is comparable for the two forecasting chains, however, peak-box uncertainty for DRP-CE is again considerably larger.

## 6  Discussion

### 6.1  Expectations

In a study for the Verzasca catchment, Liechti et al. (2013a) investigated the ability of several radar-based forecasting chains for FF prediction. They found superior performance for a combination of radar ensemble with COSMO-2, referred to as REAL-C2.

In a similar analysis, Addor et al. (2011) set up forecasting chains based on PREVAH hydrological model, FLORIS hydraulic model and meteorological input from COSMO-7 and COSMO-LEPS to assess flood risk for the city of Zurich. With regard to





skill scores, both studies found (1) a decrease of skill with increasing lead time, (2) an increase in uncertainty of skill scores with increasing lead time, (3) a decrease in skill with increasing threshold quantile and (4) an increase in uncertainty of skill scores with increasing threshold quantile.

These findings reflect the physical nature of FFs: the more extreme and the further in future events are, the more difficult it is to correctly predict them (Addor et al., 2011). Therefore, these characteristics are expected to hold as well for this study, and are discussed below. Any discrepancies may reveal sampling issues and an insufficient amount of data used in this investigation.

**(1)** A clear **decrease of skill with increasing lead time** is not found for any investigated skill score. However, a tendency to decrease although there is not much change over lead time is found in POD, FAR and FB. ROCa values stay relatively constant with lead time, except for the highest thresholds where there is a clear decrease. For high threshold quantiles, there is also a substantial lowering of BSS with lead time, which is not observed for low quantiles.

**(2)** Although an **increase of uncertainty in skill scores with lead time** is expected according to Liechti et al. (2013a), it is not always visible in their graphs, e.g. in the temporal evolution of BSS. Furthermore, they found a decrease of ensemble spread from REAL-C2 in the Verzasca basin with lead time, possibly due to the nature of events included in their analysis. In this study, there is no tendency for a temporal increase in uncertainty of skill scores visible. To what extent this is due to a possible violation of the independence assumption during the bootstrapping procedure or due to the characteristics of the investigated events was not assessed.

**(3)** A **decrease of skill with increasing threshold quantile** is explicitly present in POD and FB. However, it is not visible in BSS, as the values are highly varying with threshold quantiles. Furthermore, there are sudden "jumps" in skill in terms of BSS, e.g. for 6 hours lead time and $q_{0.95}$ quantile for the two process-based forecasting chains. For extreme events, i.e. higher thresholds, FAR values of all approaches are lower, which is in contrast to the expectations. Considering the values of ROCa, there is no clear pattern visible indicating a decrease of skill with increasing threshold.

**(4)** As in the studies from Addor et al. (2011) and Liechti et al. (2013a), there is an **increase of uncertainty with increasing threshold quantile** visible in BSS.

## 6.2 Effect of using knowledge on DRP when comparing with operational benchmark forecast

The event discussed in Fig. 9 and 10 reveals that the process-based forecasting chains are able to react faster on precipitation input than the traditional forecast. This is to some extent due to the pre-moistening phase of the traditional PREVAH forecast, which means that soil moisture storage content must rise before strong peaks in runoff are simulated. As a consequence, DRP-CE performs better than TRAD-CE in terms of peak timing ($D_{TIME}$) in the Verzasca basin. Furthermore, the process-based forecasting chains react more intense on rainfall input, leading to higher peaks in runoff but also larger uncertainties for the ensemble approach. Although the use of information about DRP decreases the hydrological model parameter uncertainty, as found by Antonetti et al. (2016b), it does not decrease the total uncertainty in forecast mode.

The skill scores support the findings from the visual inspection of the events. In general, traditional forecasts underestimate



runoff and are conservative in terms of threshold exceedances, with strong under-forecasting especially for high quantiles in both catchments (Fig. 6, lower panel). For process-based forecasting chains, the opposite is true: there is in general an overestimation of runoff and an over-forecasting of threshold exceedances. This results in a higher POD for the process-based forecasts at the cost of a relatively high FAR.

In terms of ROCa, a striking preference for DRP-CE over TRAD-CE for high threshold quantiles relevant for FFs in both catchments and for all lead times is found. A tendency to favour the process-based forecasting chain at high quantiles is also present when investigating BSS, especially for short lead times. In some cases, e.g. for $q_{0.8}$ and $q_{0.95}$ quantiles in Verzasca and $q_{0.95}$ and $q_{0.975}$ quantiles in Pincascia basin, process-based forecasting chain outperforms the traditional forecast significantly with uncertainty bars not overlapping (Fig. 7). However, it is not easy to state whether this "starting gap" is due to the usage of

RGM-PRO instead of tradtional PREVAH or to the inclusion of CombiPrecip instead of pluviometer data. In general, overall performances in terms of BSS are highly dependent on chosen threshold quantile and lead time.

In contrast to the anticipation of Antonetti et al. (2016a), process-based forecasts are not of more advantage in nested Pincascia basin. The traditional forecast is more competitive in terms of ROCa and also BSS there. In particular for the peak-box approach in Pincascia basin, IQR hit rate is substantially lower for DRP-CE although both peak-box and IQR boxes are

on average much larger. In contrast, hit rates are at least comparable between the two models in the Verzasca catchment. However, there is still the disadvantage for the process-based forecasting chain of having larger uncertainties due to its fast reacting features. For further investigations of the peak-box approach it might be insightful to separate uncertainty of timing and magnitude.

### 6.3   Comparison with Liechti et al. (2013a)

In the study of Liechti et al. (2013a), all forecasting chains revealed POD values higher than 0.8 on all thresholds for a lead time of 6 hours. This holds as well for DRP-C1 and median of DRP-CE in this study, but not for the traditional forecasts. TRAD-C1 and median of TRAD-CE have a substantially lower POD, i.e. around 0.5 for highest quantiles. In the investigation of Liechti et al. (2013a), FAR values increased with higher threshold quantiles from 0.15 to around 0.4. In contrast, process-based forecasts of this study start with FAR values of 0.2 for $q_{0.5}$ threshold quantile, rise to values higher than 0.3 for $q_{0.7}$ and

then drop down towards zero for $q_{0.95}$ threshold quantile. Values of FAR for traditional forecasts TRAD-C1 and median of TRAD-CE are close to zero for medium to high quantiles. In terms of FB, best performing forecasting chain of Liechti et al. (2013a), REAL-C2, revealed over-forecasting of large quantiles with a similiar magnitude as the two process-based forecasting chains in this study. Contrary to this, traditional forecasts TRAD-C1 and median of TRAD-CE show strong under-forecasting for all threshold quantiles. The use of a logarithmic scale to display FB would be preferential because there is perspective distortion for a linear scale. However, it was not done in order to assure comparability with Liechti et al. (2013a). Considering

BSS, values of DRP-C1 and DRP-CE for a lead time of 6 hours are around 0.6 for $q_{0.8}$ and around 0.4 for $q_{0.9}$ quantile, which matches remarkably well with REAL-C2 from Liechti et al. (2013a). For traditional forecasts, BSS values are around 0.1 for $q_{0.8}$ and around 0.5 for $q_{0.9}$ quantile, where this strong sensitivity to threshold quantile could be an indicator of sampling issues as mentioned above. Similarly as in this study, ROCa showed sometimes an increase with higher threshold quantiles in Liechti



et al. (2013a). In turn, they found a decrease in ROCa values with increasing lead time, which is not very pronounced in the case of this study.

One could conclude that for high quantiles, process-based forecasting chains of this study are better than forecasts inves-
tigated in Liechti et al. (2013a), as they have a comparable POD but a much lower FAR. However, the methodology of the
two studies was too different for such a reasoning. In contrast to Liechti et al. (2013a), this study operated with windows of 6
hours for computations of POD, FAR, FB and BSS in order to have sufficient amount of data. Furthermore, time periods and
therefore included events of the two investigations differ significantly.

## 6.4  Effect of using a meteorological ensemble

In accordance with Addor et al. (2011), Liechti et al. (2013a) and others, a clear preference for the probabilistic approaches
in both catchments and for all forecasting chains is found. In terms of spatial resolution, forecasts based on COSMO-1 with
a mesh size of 1.1 km, in comparison with the ones relying on COSMO-E having a resolution of only 2.2 km, should be
favoured. This is due to the fact that a smaller grid size allows for better representation of convective systems responsible for
FFs (Collier, 2007; MeteoSwiss, 2016). However, our results show that, despite the coarser spatial resolution, probabilistic
forecasts are to be preferred over deterministic forecasts when tackling meteorological uncertainty. This is shown by the higher
BSS values of the ensemble approaches when compared with their respective deterministic counterparts for all investigated
lead times. Furthermore, there is stronger decrease in skill with increasing lead time for deterministic than for probabilistic
approaches, which supports findings of Addor et al. (2011). Deterministic forecasting chains are most competitive for very
short lead times. This is also due to the fact that the skill of probabilistic prediction systems is not always highest for shortest
lead time but sometimes later, especially when the forecast is started from a single initial state and therefore needs time to
develop some spread.

When comparing the median of probabilistic forecasts with deterministic predictions, Ehret (2011) found a preference for
the median in ensembles for catchment-averaged rainfall. Similarly, Addor et al. (2011) showed that the median in hydrographs
driven by COSMO-LEPS outperformed deterministic forecasts from COSMO-7 in their study. In this study, no clear preference
is found for probabilistic predictions when turned into deterministic forecasts. In general, uncertainty in skill scores resulting
from bootstrapping is larger for approaches relying on COSMO-E than for the ones based on COSMO-1 as there are eight
deterministic weather forecasts but only two probabilistic predictions each day. As a consequence, there is four times less data
for the ensemble forecasting chains at a specific lead time. To strictly assess the effect of spatial NWP resolution, deterministic
skill scores for one random member of the COSMO-E hydrograph ensemble could be compared with simulations relying on
COSMO-1.

In accordance with Addor et al. (2011), this study reveals overconfidence of ensemble forecasting chains. This is the case
for DRP-CE and TRAD-CE in both Verzasca and Pincascia catchments and is visible in the verification rank histograms in the
supplementary materials. Overconfidence mainly results from situations where no ensemble member predicts any precipitation:
there is zero spread but simulated hydrographs do not match the base flow. This usually happens in low flow periods and is not
really of relevance for FF prediction.




## 6.5 Limitations

As a first limitation of this study, one has to be aware that not only FFs are investigated but also heavy runoff events that develop over days, which is also the case in the study of e.g. Liechti et al. (2013a). To treat FFs that evolve within minutes, which is part of the definition by Norbiato et al. (2008), a temporal resolution of one hour is not enough. In addition, Addor et al. (2011)
and Liechti et al. (2013a) stated that results for high threshold quantiles should be - if at all - only carefully interpreted as data is sparse. Therefore, Liechti et al. (2013a) made general conclusions only up to the $q_{0.8}$ threshold quantile. However, question arises then how relevant these quantiles are for FF prediction.

Some aspects of the skill score evaluation, i.e. the comparison of the results of this study with the expectations due to the physical nature of FFs in Sect. 6.1, revealed data sampling issues. This is for instance the case when the forecast performance
is increasing with larger threshold quantiles or longer lead times.

The findings are further limited by the occurrence of compensation problems, because a meteorological and a hydrological model were connected in series. As the goal is to have models that are right for the right reason (Seibert and McDonnell, 2002; Klemeš, 1986; Kirchner, 2006), each model would have to be evaluated separately. In particular, a quantitative analysis for the performance of COSMO-1 and COSMO-E models for precipitation predictions in the regions of interest would be desirable.
With detailed knowledge on error structure, meteorological forecasts could be pre-processed before they serve as input for the hydrological model and simulated runoff could be post-processed (Cloke and Pappenberger, 2009; Bogner et al., 2016). Furthermore, a multi-model approach (Velazquez et al., 2011) could be very interesting as the novel forecasting systems react relatively intense on precipitation whereas the traditional prediction chains are more conservative.

For the map of RTs in Verzasca region (Fig. 1), which was derived using the simplified methodology of Müller et al. (2009),
there is some potential for improvement. The fact that small patches of fast areas appear within slower regions represents an unrealistic feature, as re-infiltration would happen. This could be avoided by either applying a filter or with more expert knowledge and field work (Scherrer and Naef, 2003).

## 6.6 Synthesis of companion papers

In the companion paper (Antonetti et al., 2018), the potential of RGM-PRO in FF forecasting was assessed for the Emme basin
in the Swiss Prealps. Furthermore, the sensitivity of the predictive power to various approaches for including knowledge on runoff processes into the hydrological model, i.e. different maps of RTs, was investigated. As the Müller mapping approach was used for the setup in the Verzasca region, the forecasting chains DRP-mu-C1/CE applied in the Emme catchments are compared with the corresponding counterparts DRP-C1/CE from the Verzasca areas. A complete overview of the plots with the skill scores mentioned here can be found in the supplementary materials of the two papers.
The event case studies show that both Emme and Verzasca basins and corresponding subcatchments are prone to rapidly developing runoff peaks. However, in terms of FF prediction there is more skill for DRP-C1/CE in Verzasca basin than for the corresponding DRP-mu-C1/CE in Emme catchments. Values in Nash–Sutcliffe efficiency (NSE) and Kling–Gupta efficiency (KGE) from DRP-C1 are higher than the ones from DRP-mu-C1 and the decrease with lead time is less pronounced. In terms



of POD, FAR and FB, the performance in the Emme and Verzasca region is comparable. Considering BSS, there is much more skill for high quantiles and long lead times in the Verzasca catchments. Furthermore, values of ROCa show a strong temporal decrease for the Emme catchments but are relatively constant over time for Verzasca basins. In general, this study finds more skill especially for large event intensities and long lead times in the Verzasca catchments in comparison with the Emme basins.

A reason why the performance of the forecasting chains is superior in the Verzasca catchments could be that the Verzasca basins are more topography driven with more shallow soils and smaller dependence on initial conditions (Zappa et al., 2011).

Although DRP-C1/CE and DRP-mu-C1/CE are built with the same components, one has to be aware that evaluation differs between the two regions in terms of RGM-PRO activation. It remains unclear after this study to what extent the skill scores, their uncertainty and their variability with lead time and threshold quantiles are influenced by only taking a subset of events

with RGM-PRO activation.

## 7   Conclusions

This paper assessed the potential of a newly developed process-based runoff generation module for operational flash flood forecasting in the Verzasca basin and the Pincascia subcatchment in the southern Swiss Alps. Two quasi-operational forecasting chains were set up, including (a) numerical weather prediction data as meteorological input from deterministic COSMO-1

(1.1 km grid resolution) and probabilistic COSMO-E (2.2 km mesh size, 21 ensemble members) respectively, (b) RGM-PRO, a conceptual hydrological module with no need for calibration (Antonetti et al., 2016a), (c) gridded precipitation nowcasts from CombiPrecip, which also served as meteorological input for the reference runs (see supplementary material), and (d) operationally available soil moisture data estimated by the PREVAH hydrological model.

For the investigated region, an operational flash flood prediction system run by the Swiss Federal Institute for Forest, Snow

and Landscape Research (WSL) already exists, namely a combination of the traditional PREVAH with COSMO-1/COSMO-E based on initialisation with pluviometer data (TRAD-C1/TRAD-CE). In this area, RGM-PRO was set up with Müller mapping approach to include knowledge on dominant runoff processes into hydrological model, CombiPrecip data for initialisation and COSMO-1/COSMO-E precipitation predictions (DRP-C1/DRP-CE). This setup allowed an evaluation of the competitiveness and possible benefits of a hydrological module including knowledge on runoff processes in comparison with an operational

benchmark prediction system. Predictive power of the two systems was assessed on 22 specific days from May to August 2016.

> Results indicate that the forecasting chains including information on runoff processes reacted faster and more intense to precipitation input when compared with the operational benchmark forecasting chains. This led to larger spread for hydrological ensemble predictions with DRP-CE. As a further consequence, DRP-C1 and DRP-CE median had

higher values of probability of detection than TRAD-C1 and TRAD-CE median at the cost of a larger false alarm ratio. Considering skill in terms of the area under a ROC curve, we found a striking preference for the new forecasting system for high threshold quantiles relevant for flash flood prediction. In both Verzasca and Pincascia basins, the novel forecasting chains were competitive with the operational benchmark forecasts.





Furthermore, for all catchments, the vast majorities of event intensities and for all lead times, we found a clear superiority of forecasting chains including a meteorological ensemble. This supports findings from Addor et al. (2011) and Liechti et al. (2013a) and highlights the importance of accounting for uncertainty in location, timing and intensity of precipitation (Addor et al., 2011; Liechti et al., 2013a).

We are aware that studying extreme events with only one season of data is not fully appropriate. Therefore, we will repeat the statistical analysis after several years of experience with the new forecasting chains. However, the main work was the setup of the novel module in quasi-operational use, and to evaluate its potential compared to the pre-existing forecasting chains.

In general, it can be concluded that the newly developed forecasting chains can compete with the traditional prediction systems in gauged catchments. This is remarkable, as the traditional systems rely on long-term runoff measurements for the calibration of the hydrological model. With the new forecasting chains, successful expansion of operational flash flood prediction to ungauged basins should be possible. Necessary is the availability of a digital terrain model and spatial information on geology and land use.

Although there is already skill in flash flood forecasting for many event intensities and lead times, further understanding of flash floods as a natural hazard is needed for model development and to improve predictions. With this, effective measures for civil protection are enabled.

## 8  Data availability

Arealstatistik 1979/85 (http://www.bfs.admin.ch) with 100 m resolution is applied as land use map and DTM25 (Data: BFS GEOSTAT/Federal Office of Topography swisstopo, http://www.swisstopo.admin.ch) with a resolution of 25 m as digital terrain model for the Müller map needed for RGM-PRO. Geologischer Atlas GA25 (Data: BFS GEOSTAT/Federal Office of Topography swisstopo) with a scale of 1:25'000 is used where it is available and elsewhere Geologische Karte (Data: BFS GEOSTAT/Federal Office of Topography swisstopo) with a scale of 1:500'000 is used as geological map. Meteorological input data, i.e. CombiPrecip, COSMO and rain gauge data, is obtained from the Swiss Federal Office of Meteorology and Climatology (MeteoSwiss, http://www.meteoswiss.admin.ch). Runoff measurements are provided by Swiss Federal Office for Environment (FOEN, http://www.bafu.admin.ch).

*Author contributions.* Massimiliano Zappa and Manuel Antonetti designed this study. Katharina Liechti supervised and collected the operational runs of the traditional PREVAH-HRU model for the Verzasca. Massimiliano Zappa set up the COSMO information for use in PREVAH-HRU and RGM-PRO. The RGM-PRO hindcasts have been completed by Manuel Antonetti and Christoph Horat. The statistical analysis was carried out by Christoph Horat using R scripts originally designed by Katharina Liechti. The verification of the COSMO forecasts was completed at Swiss Federal Office of Meteorology and Climatology (MeteoSwiss) by Pirmin Kaufmann. Christoph Horat prepared the manuscript with contribution from co-authors.



*Acknowledgements.* The contribution of Manuel Antonetti has been funded by the Swiss Federal Office for Environment (FOEN). Combi-Precip data have been prepared by Dr. Ioannis Sideris from Swiss Federal Office of Meteorology and Climatology (MeteoSwiss). We are grateful to Prof. Heini Wernli (ETH Zurich) for supervising the thesis of Christoph Horat and for his valuable comments on parts of the manuscript.



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

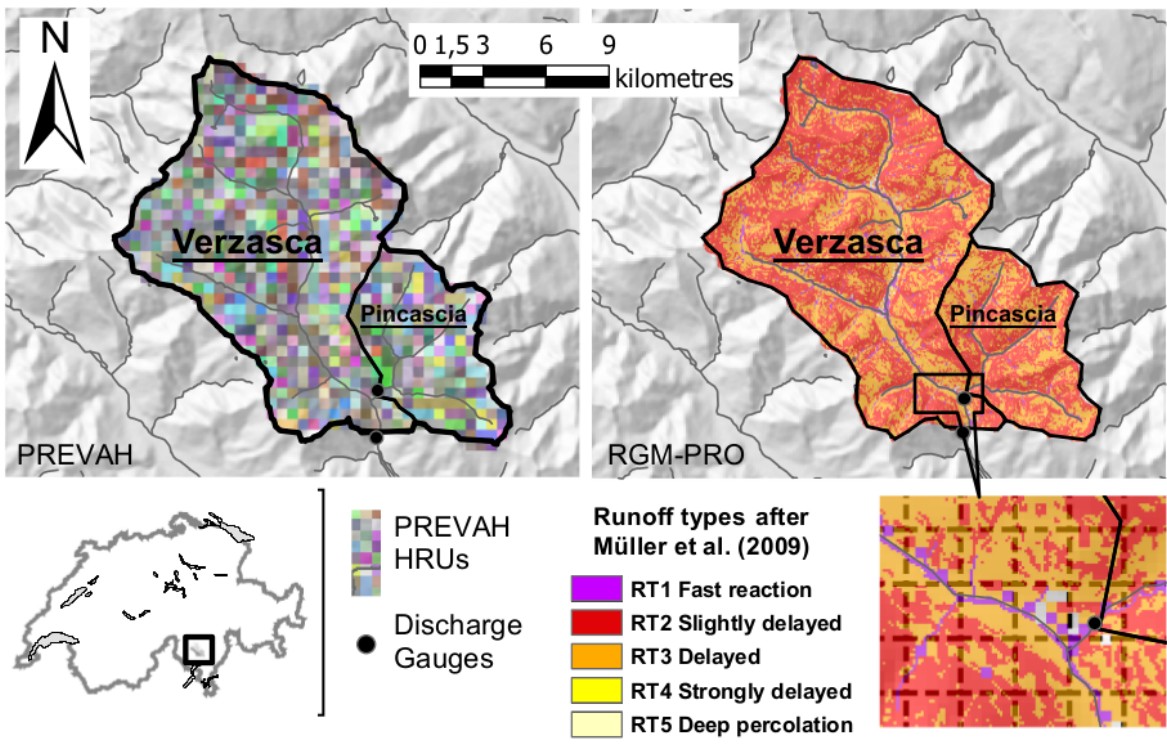

**Figure 1.** Verzasca basin and Pincascia subcatchment with discharge gauges indicated and PREVAH HRUs on the left and RTs as used in RGM-PRO after Müller et al. (2009) on the right hand side. The grey pixels in the map of RTs denote built up areas.





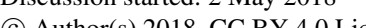

**Figure 2.** Scheme of the FF forecasting chains in the Verzasca catchments investigated in this study with keys indicated on the right hand side. On the left hand side, P.I. stands for *precipitation input*, H.M. for *hydrological model* and S.D. for *soil moisture data*.





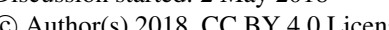

**Figure 3.** Comparison of COSMO-1 and COSMO-2 model performance in terms of POD, FAR (upper panel) and FB (lower panel) for 12 hourly accumulated precipitation and a lead time of 13-24 hours as a function of threshold in summer season (s3, JJA) 2016.





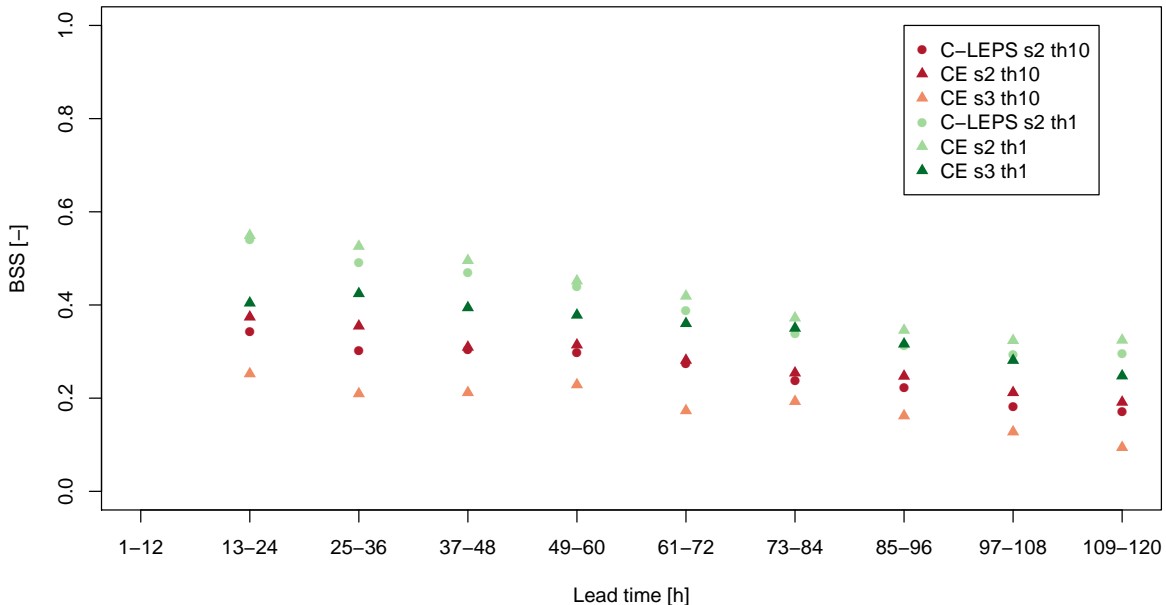

**Figure 4.** Comparison of COSMO-E and COSMO-LEPS model performance in terms of BSS for spring (s2, MAM) and summer (s3, JJA) season for precipitation threshold exceedance of 1 mm (th1) and 10 mm (th10) as a function of lead time.

**Table 1.** Summary of peak-box results for Verzasca and Pincascia catchments for DRP-CE and TRAD-CE.

|  | Verzasca | | Pincascia | |
| --- | --- | --- | --- | --- |
|  | **DRP-CE** | **TRAD-CE** | **DRP-CE** | **TRAD-CE** |
| $D_{PEAK}$ (median) [m$^3$/s] | 25 | 19 | 7 | 6 |
| $D_{TIME}$ (median) [h] | 3 | 7 | 8 | 7 |
| IQR hit [%] | 11 | 9 | 2 | 11 |
| Peak-box (and no IQR) hit [%] | 52 | 50 | 34 | 36 |
| no hit [%] | 36 | 41 | 64 | 52 |
| $UC_{IQR-box}$ (median) [mm] | 7 | 4 | 11 | 3 |
| $UC_{peak-box}$ (median) [mm] | 121 | 84 | 133 | 92 |




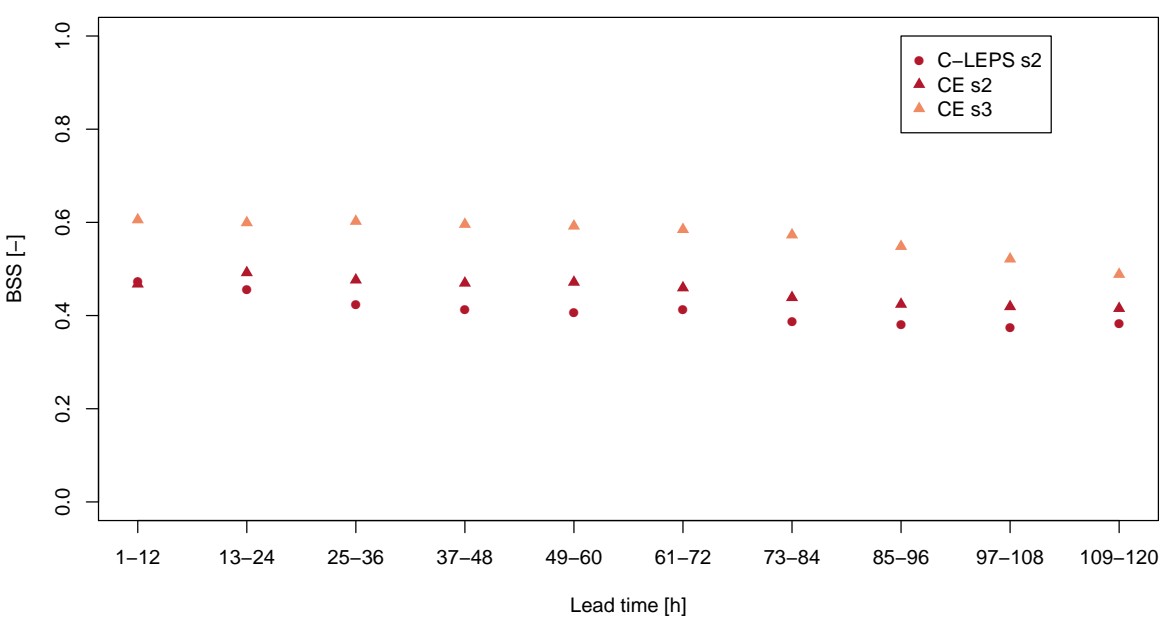

**Figure 5.** Comparison of COSMO-E and COSMO-LEPS model performance in terms of BSS for spring (s2, MAM) and summer (s3, JJA) season for temperature threshold exceedance of 20 °C as a function of lead time.





**Figure 6.** POD, FAR (upper panel) and FB (lower panel) for Verzasca catchment as a function of threshold quantile and for several lead times for DRP-C1, TRAD-C1, DRP-CE (med) and TRAD-CE (med). A window of 6 hours was taken for the computations, e.g. values from 19 h to 24 h were considered for the 24 h lead time.




**Figure 7.** Comparison of BSS in Verzasca catchment for deterministic DRP-C1 and TRAD-C1 and probabilistic DRP-CE and TRAD-CE as a function of lead time for several threshold quantiles. A window of 6 hours was taken for the computations, e.g. from 19 h to 24 h for the 24 h lead time. The boxplots represent the sampling uncertainties of the score computations obtained with bootstrapping.





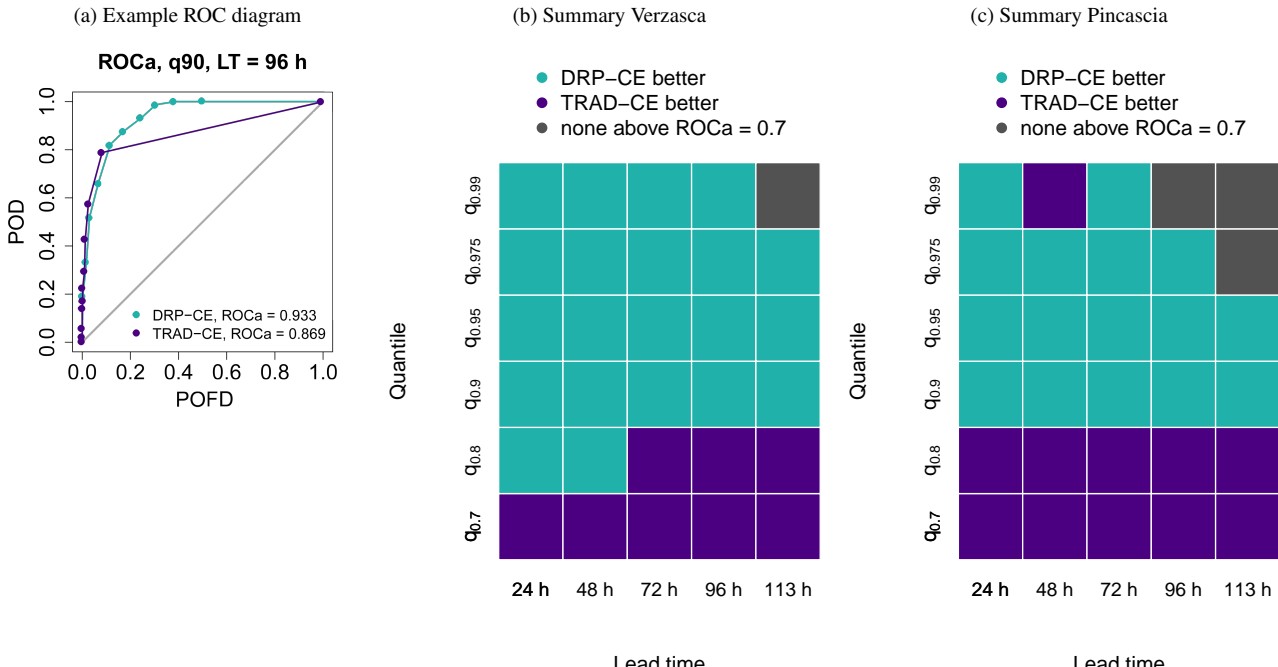

**Figure 8.** Summaries of ROCa for Verzasca (b) and Pincascia (c) as a function of lead time and threshold quantile for TRAD-CE and DRP-CE. Blue colour indicates that ROCa of DRP-CE is higher, whereas purple colour implies that TRAD-CE performs better. Grey shading indicates that none of the forecasting chains has ROCa higher than 0.7, which is considered to be the minimum value useful for decision makers (Buizza et al., 1999). Summaries are based on ROC diagrams, of which an example is shown in (a) for the Verzasca basin: ROC curve for TRAD-CE (purple) and DRP-CE (blue) are indicated for a lead time of 96 hours and $q_{0.9}$ threshold quantile with corresponding ROCa. Please note that steps in probability thresholds of 0.1 are used. A window of 24 hours was taken for the computations, e.g. values from 25 h to 48 h were considered for the 48 h lead time.



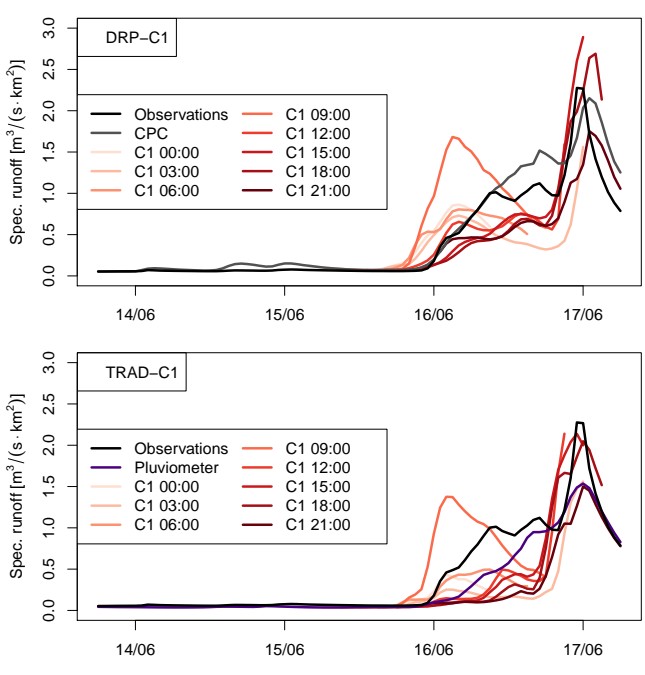

**Figure 9.** Deterministic flood prediction with DRP-C1 (top) and TRAD-C1 (bottom panel) in Verzasca basin, namely time-lagged ensemble of eight hydrographs forced with different COSMO-1 runs from $15^{th}$ of June, 2016.

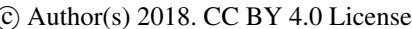




**Figure 10.** Ensemble flood prediction for the largest event in Verzasca basin investigated in this study with switch to forecast mode at 19:00 on $15^{th}$ of June, 2016. Probabilistic precipitiation forecasts from COSMO-E and comparison with CombiPrecip is shown in top panel. Second and fourth panel depict ensemble area plots and third and fifth panel show peak-box approach for DRP-CE and TRAD-CE, respectively.