# Peer review of "S1 Runoff threshold quantiles"

_Natural Hazards and Earth System Sciences, 2018_

## Referee Comment (RC1) · E. Gaume (Referee) · 20 Jul 2018

The manuscript submitted by Horat and al. compares a newly developed flash flood forecasting chain based on a so-called "process-based" rainfall-runoff model to a chain that is running operationally in Switzerland. Some additional elements of comparison of the respective qualities of deterministic and probabilistic forecasts are proposed (i.e. this question was the main issue of previous publications of the same authors). The paper builds on the large experience of the authors in the field of flood and especially flash flood forecasting. The methods are clear and well-established except for the use

of the Brier score to compare deterministic and probabilistic forecasts (see hereafter). The connection with real-world operational methods and application is a very positive aspect of the presented work. To my knowledge, particularly advanced methods are implemented operationally in Switzerland if compared to the rest of the world. Nevertheless, the proposed comparison is conducted for one single watershed (and one of its sub-watersheds but the results are not presented in the manuscript) and for a period of time of 3 months only (May to August 2016). This is by far insufficient to draw real convincing conclusions. The implementation of a rainfall-runoff model without calibration can only be evaluated if conducted on a significant number of watersheds – typically some tens. Likewise, a three months period seems far to short for a faithful evaluation of forecasting models and does not correspond to the general standards of the scientific publications. The results may be too dependent on some few if not a single flood event with no possibility of generalization. The authors themselves acknowledge in the discussion part of their manuscript that sparse data may be problematic (P 18 L6). They also mention that the obtained results and skills and their variations with lead times or considered threshold quantiles are not consistent with the ones obtained in previous studies. There is, to my opinion, a high probability that part of these observed inconsistencies may be explained by the limited size of the test data set. At least, the authors have to demonstrate the robustness of their results and interpretations. The objectives and methods presented in the paper are correct, but the manuscript can hardly be published to my opinion unless a much larger data set in time and space (larger period of time and larger number of watershed is considered). The team seems to have access to reach data sets in Switzerland ; It is time consuming of course, but I do not see any reason why they could not conduct a large and necessary test and validation study based on the approach presented in the manuscript. Apart from this major issue, some other less important comments can be done on the presented manuscript:

1) The authors mostly refer to their own works. Indeed, interesting and innovative methods are implemented in Switzerland to forecast flash floods. But it would important also to cite works conducted in other countries and by other teams on the

same issue at least in the introduction of the manuscript to show the originality of the proposed approach. Flash flood forecasting has been an active field of research in the recent years. 2) The manuscript refers in many places to a companion paper and to supplementary materials. This is frustrating for the readers since some important information is not provided in the manuscript such as the implementation of the "process based" model (what are the input variables, how are the values of the parameters of the model fixed) or the results obtained for the Pincascia sub-watershed. Supplementary material is interesting but a manuscript must be to a certain extent self-sufficient and contain at least the basic information needed for the interpretations and the results that are commented and interpreted. 3) Brier scores are used to compare deterministic and probabilistic forecasts. I know that some other papers did the same, but this comparison is not appropriate. Indeed, a Brier score can be computed in both cases, but do not measure exactly the same things and can therefore not be directly compared. Forecasts must be combined with a utility function and evaluated in a decision making context for a proper and rigorous comparison. An annotated manuscript is attached to this review.

Please also note the supplement to this comment:
https://www.nat-hazards-earth-syst-sci-discuss.net/nhess-2018-119/nhess-2018-119-RC1-supplement.pdf

---

## Referee Comment (RC2) · Anonymous Referee #2 · 3 Aug 2018

Overview

The manuscript presents an application of an event-based runoff generation module driven by forecasts provided at high horizontal resolution by two new numerical weather prediction tools (a deterministic one and a probabilistic one). The meteo-hydrological model coupling is tested for a small-size catchment (and its sub-catchment) in a mountainous area of Switzerland. The manuscript addresses relevant and interesting methods (in particular, the proposed hydrological module does not need a calibration task) to improve flood forecasting. The description of the experiments, calculations and re-

sults is clear and accurate. However, the results are based on a very limited dataset (just one summer season for the meteorological analysis and about twenty events for the hydrological analysis), that may result as not sufficient to support the interpretations and the conclusions.

General comments

(1) It is not so clear the main goal of the manuscript, with respect to the companion paper. On the one hand, if the focus of the paper is on the meteorological input (as stated by authors), therefore the dataset may results as quite limited to test the improvements of the new meteorological forecasting tools (why have authors not considered a larger period of data availability of COSMO-1 and COSMO-E? For instance, the years 2016-2018?). Actually, if the focus of the paper is the evaluation of the performance provided by the new meteorological chains, then the computation of the statistical scores could be carried out over a longer period, without the need to perform a comparison with the older forecasting chains (model benchmarking), given that the statistical scores are able to give an objective evaluation of performance for the tested meteorological forecasting tools. On the other hand, if the focus of the research is on the performance of the new meteo-hydrological chain, therefore the dataset seems to be not so significance in terms of flood events, in particular with respect to the operational aims of civil protection authorities (just one event with a 2-yr return period in the investigated dataset). The limited amount of data here used for the statistical analysis may not justify a separate manuscript with respect to the companion paper. Maybe, the investigations and results shown in the present manuscript could be synthesized (as done, for instance, in Section 6.6) and added to the companion paper in order to enlarge the statistics for the proposed coupling of RGM-PRO approach to COSMO-1 and COSMO-E.

(2) The meteo-hydrological model coupling is used as a verification tool for the new meteorological model chains at higher resolution. But, by the hydrological perspective, it seems that there are not enough high-impact events in the investigated period. Moreover, it could be questionable that the false alarms are "realistically" evaluated (Pag.8, lines 7-9), given that the investigated period is the summer season. I mean, which is the soil saturation of the study area in summer? Is summer a dry season for the study area or the soil saturation is quite high in summer so that a light/moderate rain event could trigger a flood event? Or are there floods in summer only due to extreme rainfall events which cause rapid surface runoff without rainfall infiltration? Authors should add "hydrological" details about the occurrence of flood events in summer for the selected catchments.

(3) The introduction (i.e., Section 1) could be shortened (for instance, lines 14-33 at Pag.2; lines 13-34 at Pag.3; lines 1-31 at Pag.4). Some issues are repeatedly discussed and too much detailed descriptions of past studies are provided, even though not strictly related to the contents and methodologies proposed in the manuscript. Thus, a synthesis may result advantageous. Moreover, the contents may appear as dispersive (too general) with respect to the context of the study area and the proposed forecasting methodologies. The contents of Section 1 should focus on contents which show similar features to the present study. The citation of past studies should highlight the feasibility of those approaches with respect to spatial and temporal characteristics of phenomena (for instance, catchment dimension, return time of the basin, forecast lead time), focusing on the similarities with the present manuscript. In the current form, this section seems as a general review of the flood forecasting subject.

(4) I guess that the hourly runoff climatology of the period May-August 2016 was used as reference climatology to carry out the statistical analyses (for instance, to compute the quantiles of Figures 6-8). Is the May-August 2016 runoff climatology statistically meaningful with respect to a longer climatology (for instance, some decades) for the selected study area? Section 5 provides a very detailed analysis of the performance for the tested forecasting chains. Nevertheless, it is not so evident that these performances are significantly different. Even, some scores provide outcomes in contrast to the companion paper (for instance, the process-based forecasts are not better for the

nested sub-catchment). The limited dataset may hamper a solid comparison.

(5) Sections from 6.3 to 6.6 go into a detailed analysis about comparisons of the proposed new forecasting chains with previous studies. However, models, data input, study areas and investigated period are not always the same. Therefore, the content of these sections may result too long, redundant and not so interesting with respect to evaluation of the proposed forecasting chains. The discussion recalls results and trends which are general and well known in the past specific literature (in particular Section 6.4). Authors should highlight the original contribution of the proposed forecasting chains and summarize the comments on the comparisons (for instance, authors could move each specific comments of Section 6.5 in a position within the manuscript where that issue has already been discussed, rather than devote a specific section to comment all the issues of the comparisons).

Specific comments

Pag.1, Lines 20-22: The comment on the performance of the proposed model chains should stress the feasibility of these chains with respect to the dimension of the study area. This point of view for the discussion of results could be an added value for the present manuscript.

Pag.2, Lines 1-2: This statement should be based on a larger dataset.

Pag.2, Line 11: The meteorological perspective is not so deeply investigated in the manuscript.

Pag.3, Line 2: Authors should specify the reasons for the suitability just for catchments with areas up to 1000-2000 km2.

Pag.3, Line 18: Authors should specify the country of FOEN.

Pag.3, Line 29: Authors should specify the year of the event.

Pag.6, Line 10: Authors should specify the meaning of the acronym "WSL".

Pag.6, Line 10: Authors should specify that PREVAH is a semi-distributed hydrological model (as done in the abstract).

Pag.7, Line 9: Should "and Avalanche Research SLF." be "and Avalanche Research (SLF)."?

Pag.7, Lines 15-22: Is the configuration of the COSMO model changed with the increase of horizontal resolution (from 2.2 to 1.1 km)? Authors should add details and references about this issue.

Pag.7, Lines 23-27: Is the configuration of the COSMO-based ensemble changed with the increase of horizontal resolution (from 7 km of COSMO-LEPS to 2.2 km of COSMO-E)? Authors should add details and references about this issue.

Pag.8, Lines 1-3: This sentence is not clear.

Pag.8, Lines 6-7: Does a threshold exist to identify major flood events (namely, flood events which are of interest for the authority in charge of the public safety)? How many major flood events occurred in summer 2016 for the study area?

Pag.8, Lines 23-24: The comparison may results as not fully proper, given that the observed rainfall input is different for the two chains. Why has the same input not been used for both the chains?

Pag.9, Line 13: Is the hourly runoff climatology of the period May-August 2016 statistically meaningful with respect to a longer climatology (for instance, some decades) for the selected study area? It could be useful to show a comparison between the reference climatology of this study and a historical ones for the selected catchment.

Pag.9, Lines 14-16: Are the statistical scores computed at each hourly time step of the simulation event? I mean, is the threshold exceedance evaluated each hour and the corresponding score computed at an hourly time step, then averaged over each lead time window? Or is the threshold exceedance evaluated just one time within the whole lead time window? Please clarify.

[Figure]

Pag.10, Lines 10-13: The description of the Brier Score decomposition could be omitted, given that it is not discussed in the main manuscript.

Pag.11, Lines 20-21: Which is the spatial domain (Switzerland? Verzasca catchment?) over which the scores shown in Fig.3 were computed? Please specify. The scores for POD and FAR are not so satisfying, especially for the higher thresholds (namely, rainfall events which likely trigger flood peaks). The FAR scores may results quite high with respect to the usefulness for operational decisions of civil protection authorities. Could authors add a comment to justify this results? Which is the rainfall threshold that trigger major flood events for the study catchments?

Pag.12, Lines 1-2: Why are the BSS values not shown in Fig.4 (or discussed) for the thresholds higher than 10 mm (as done in Fig.3)?

Pag.12, Lines 14-15: Please specify that the cited scores refer to runoff data and the quantiles refer to the hourly runoff climatology of summer 2016 (in case of my interpretation is right).

Pag.13, Line 14: The panels "b" and "c" of Fig.8 are very friendly to convey the best performing method, but this visualization does not allow to evaluate if the difference of performance is significant in term of ROCa.

Pag.13, Lines 16-19: Authors should try to justify this result. May the reason for this result be ascribed to the larger dimension of the Verzasca catchment (with respect to its sub-catchment), which can allow to "average" spatial errors in the localization of the rainfall field?

Pag.13, Lines 21-22: The magnitude of the event (i.e., 2-yr return period) should also be cited here (as done at Pag.14, Line 4). Authors could add a comment about the frequency and amount of the flood peaks corresponding to more intense events for the selected catchments.

Pag.13, Lines 24-30: The significance of the comparison may result as weak, given

that it is discussed for two forecasting chains that differ for the model and input used. Authors should add a comment in order to justify this result with respect to model characteristics and data input.

Pag.15, Line 6: With this statement authors recognize that the discussed results and conclusions may be invalidated by the limited dataset that has been used to test the proposed new forecasting chain. Actually, the usefulness and added value of the new forecasting chain are questionable due to the limited test period. The use of a different dataset could provide different (and opposing) results.

Pag.16, Lines 1-19: The detailed comment on the comparison results does not lead to a clear conclusion about which chain should be preferable. Some scores provide opposing outcomes (for instance, in contrast to the companion paper, the process-based forecasts are not better in the sub-catchment), then this analysis may result as inconclusive.

Pag.19, Line 25: Authors should specify the magnitude of these 22 events with respect to the catchment climatology.

Pag.20, Lines 1-4: This result is quite general and recalls several past studies. Here, it appears as a local application based on a limited dataset.

Pag.20, Lines 5-12: Authors recognize the major drawback of the present study. They would evaluate a new model-based approach to flood forecasting which does not require the calibration task for the hydrological module, but the available dataset is not appropriate (due to the very limited size) to prove the added value of the proposed approach.

---

## Author Comment (AC1) · 3 Oct 2018

**Ensemble flood forecasting considering dominant runoff processes: II. Benchmark against a state-of-the-art model-chain (Verzasca, Switzerland)"**

Authors replies to RC2:

We want to thank the reviewer for his/her assessment of our manuscript. In the following we give our answers to the comments and recommendations that have been raised. Reviewer comments RC are **bold,** our reply AR is in *italic*. Insertions in the revised manuscript MI are underlined.
* * *
**GENERAL ASSESSMENT**

**RC: … the results are based on a very limited dataset (just one summer season for the meteorological analysis and about twenty events for the hydrological analysis), that may result as not sufficient to support the interpretations and the conclusions.**

*AR: This issue has been raised also by the reviewer 1 and by the reviewers of the companion paper by Antonetti et al. (2018). We are of course aware, that more basins and longer periods of evaluation are always welcome. NHESS (but also HESS) is in this respect a journal that regularly publishes case studies (e.g. Kobayashi et al., 2016; Cane et al., 2013), preliminary assessments (Picciotti et al., 2013) or intercomparison of approaches during limited period of time (e.g. Davoli et al., 2018; Li et al., 2018). Having targeted NHESS as journal for disseminating our experience, this study is designed to benchmark a state-of-the-art and operational calibrated hydrological prediction system(PREVAH-HRU) against a newly developed event-based system that is configured without calibration requirements (RGM-PRO). This has been evaluated during a representative flood season and in case a nested basins where PREVAH-HRU simulations have been running and collected in real-time (so no tailored experiment here, 100% data from a deployed system), while RGM-PRO simulations have been completed as reforecast experiment in the framework of a master project (August 2016 to February 2017) by the lead author. With this approach we can learn about the quality of the novel approaches from different perspectives at the same time. The transfer of experience to another catchment and climatic region is presented in the companion paper by Antonetti et al. (2018), while in prior studies we shown how PREVAH-HRU behave in case of long-term analyses (e.g. Addor et al., 2011 and Zappa et al., 2011). This is al why we take so much time and space in order to discuss these findings with respect to our previous studies. Furthermore, As far as the length of the investigation period is concerned, some limitations arise from the use of COSMO-E and COSMO-1. MeteoSwiss decommissioned after several years the antecedent operational NWP COSMO-2 and COSMO-LEPS in 2016. As we want to make our systems operational, it was for us important to focus on a first analysis with the new NWPS that we receive and archive in real-time since February 2016.*

*In the revised manuscript we will better declare our choices concerning selection of basins and investigation period. Furthermore we will try to indicate to which extent the available season is representative with respect to log-term discharge observations in the target area. We have currently unfortunately no capacity to extend the analyses beyond 2016, and in case of 2018, this would not beneficial since no severe flood event occurred.*
* * *
**GENERAL COMMENTS**

**RC: (1) It is not so clear the main goal of the manuscript, with respect to the companion paper. On the one hand, if the focus of the paper is on the meteorological input (as stated by authors), therefore the dataset may results as quite limited to test the improvements of the new meteorological forecasting tools (why have authors not considered a larger period of data availability of COSMO-1 and COSMO-E? For instance, the years 2016-2018?). Actually, if the focus of the paper is the evaluation of the performance provided by the new meteorological chains, then the computation of the statistical scores could be carried out over a longer period, without the need to perform a comparison with the older forecasting chains (model benchmarking), given that the statistical scores are able to give an objective evaluation of performance for the tested meteorological forecasting tools. On the other hand, if the focus of the research is on the performance of the new meteo-hydrological chain, therefore the dataset seems to be not so significance in terms of flood events, in particular with respect to the operational aims of civil protection authorities (just one event with a 2-yr return period in the investigated dataset). The limited amount of data here used for the statistical analysis may not justify a separate manuscript with respect to the companion paper. Maybe, the investigations and results shown in the present manuscript could be synthesized (as done, for instance, in Section 6.6) and added to the companion paper in order to enlarge the statistics for the proposed coupling of RGM-PRO approach to COSMO-1 and COSMO-E.**

*AR: We thank the reviewer for this (meaningful) remark. Our original manuscript was probably not enough clear in this respect, but both companion papers focus in first order on the hydrological predictions chains evaluated. The focus on the meteorological input we declares, refers on the focus in the presentation of the data and method used As we are willing to avoid way too large overlaps between the two paper, we decided to focus the introduction of methods of paper I by Antonetti et al. on the setup of RGM-PRO and on Combiprecip and on classic measures of agreement in hydrological modelling, while in this manuscript by Horat et al. we emphasize the method section on numerical weather prediction (NWP, including a small assessment) and on the common metrics used for verification of (hydrological) ensemble predictions . We are sorry that the reviewer expected a "more meteorological" contribution and we will make our best to make our choice fully intelligible in the revised manuscript.*

*We kind of agree with the reviewer suggestion to make a synthesis of this manuscript and include it in the companion paper. We explored this option, but we have never been happy with it. As already stated, this would have implied to include the details on PREVAH-HRU, on the NWPs and on the verification metrics in the companion paper, making it way to long (and coincident with the 106 pages thesis by the lead author of this manuscript). If the editor is the opinion that we should explore a merging of the manuscripts, then we can work in this direction.*

**RC: (2) The meteo-hydrological model coupling is used as a verification tool for the new meteorological model chains at higher resolution. But, by the hydrological perspective, it seems that there are not enough high-impact events in the investigated period. Moreover, it could be questionable that the false alarms are "realistically" evaluated (Pag.8, lines 7-9), given that the investigated period is the summer season. I mean, which is the soil saturation of the study area in summer? Is summer a dry season for the study area or the soil saturation is quite high in summer so that a light/moderate rain event could trigger a flood event? Or are there floods in summer only due to extreme rainfall events which cause rapid surface runoff without rainfall infiltration? Authors should add "hydrological" details about the occurrence of flood events in summer for the selected catchments.**

*AR: We will add the required details on flood generation in the Verzasca area in summer. As the basin is in a region with steep topography the soils are quite shallow and, as discussed in Zappa et al. (2011) initial conditions are not very sensitive with respect to controlling peak discharge in summer.*

**RC: (3) The introduction (i.e., Section 1) could be shortened (for instance, lines 14-33 at Pag.2; lines 13-34 at Pag.3; lines 1-31 at Pag.4). Some issues are repeatedly discussed and too much detailed descriptions of past studies are provided, even though not strictly related to the contents and methodologies proposed in the manuscript. Thus, a synthesis may result advantageous. Moreover, the contents may appear as dispersive (too general) with respect to the context of the study area and the proposed forecasting methodologies. The contents of Section 1 should focus on contents which show similar features to the present study. The citation of past studies should highlight the feasibility of those approaches with respect to spatial and temporal characteristics of phenomena (for instance, catchment dimension, return time of the basin, forecast lead time), focusing on the similarities with the present manuscript. In the current form, this section seems as a general review of the flood forecasting subject.**

*AR: Both reviewer raise this issue. A more concise and specific introduction will be prepared.*

**RC: (4) I guess that the hourly runoff climatology of the period May-August 2016 was used as reference climatology to carry out the statistical analyses (for instance, to compute the quantiles of Figures 6-8). Is the May-August 2016 runoff climatology statistically meaningful with respect to a longer climatology (for instance, some decades) for the selected study area? Section 5 provides a very detailed analysis of the performance**

for the tested forecasting chains. Nevertheless, it is not so evident that these performances are significantly different. Even, some scores provide outcomes in contrast to the companion paper (for instance, the process-based forecasts are not better for the nested sub-catchment). The limited dataset may hamper a solid comparison.

*AR: Yes, the reviewer is correct in his assumption concerning the used "climatology". This choice and limitation due to the shot duration of the data set affects in equal ways both the RGG-PRO and the operational PREVAH-HRU chain. We are of the opinion, that even if a sound indication on the absolute quality of both chain cannot be provides, we can provide very clear indications on the difference in quality between the RGM-PRO and the benchmark (which has already been verified for longer time series using the predecessors of COSMOE and COSMO1 as forcing). For us is therefore a quite interesting and far reaching finding, that RGM-PRO can "compete" and/or "keep the pace" with a state-of-the-art system, even if no calibration is needed. We find this a very good news after years of theoretical advances on prediction of flash-floods in ungauged areas, to show that a "PUB" inspired approach yields similar quality in terms of forecasting skill in real-time mode when compared to a calibration oriented approach. Of course we can still only speculate on the results for long-term operations of such systems. Unfortunately reforecasts of such NWP are (computationally) expensive and cannot be provided.*

**RC: (5) Sections from 6.3 to 6.6 go into a detailed analysis about comparisons of the proposed new forecasting chains with previous studies. However, models, data input, study areas and investigated period are not always the same. Therefore, the content of these sections may result too long, redundant and not so interesting with respect to evaluation of the proposed forecasting chains. The discussion recalls results and trends which are general and well known in the past specific literature (in particular Section 6.4). Authors should highlight the original contribution of the proposed forecasting chains and summarize the comments on the comparisons (for instance, authors could move each specific comments of Section 6.5 in a position within the manuscript where that issue has already been discussed, rather than devote a specific section to comment all the issues of the comparisons).**

*AR: Thanks for this comment and useful suggestion that we will implement in the revised version.*
* * *
**SPECIFC COMMENTS**

**Pag.1, Lines 20-22: The comment on the performance of the proposed model chains should stress the feasibility of these chains with respect to the dimension of the study area. This point of view for the discussion of results could be an added value for the present manuscript.**

*AR: We will consider this suggestion in the revised manuscript, thanks!*

**Pag.2, Lines 1-2: This statement should be based on a larger dataset.**

*AR: The sentence now reads:* *"The findings of the two studies as obtained from a set of data limited to one flood season…"*

**Pag.2, Line 11: The meteorological perspective is not so deeply investigated in the manuscript.**

*AR: We will arrange this as replied to you general comment (1).*

**Pag.3: Line 2: Authors should specify the reasons for the suitability just for catchments with areas up to 1000-2000 km2.**

*AR: The reasons are: a) increase of chance of having more disturbed catchments. b) delineation of process maps for such large areas. c) larger catchments are not prone on the kind of flash-floods triggered by local thunderstorms, that we are focussing on. These three points will be added in the revised manuscript.*

**Pag.3, Line 18:Authors should specify the country of FOEN.**

*AR:* *Swiss Federal Office for the Environment (FOEN)*

**Pag.3, Line 29: Authors should specify the year of the event.**

*AR:* *2007*.

**Pag.6, Line 10:Authors should specify the meaning of the acronym "WSL".**

*AR:* *Swiss Federal Institute for Forest, Snow and Landscape Research*

**Pag.6, Line 10:Authors should specify that PREVAH is a semi-distributed hydrological model (as done in the abstract).**

*AR: Done.*

**Pag.7, Line 9: Should "and Avalanche Research SLF." be "and Avalanche Research (SLF)."?**

*AR: Done.*

**Pag.7, Lines 15-22: Is the configuration of the COSMO model changed with the in-crease of horizontal resolution (from 2.2 to 1.1 km)? Authors should add details and references about this issue.**

*AR: The "Numerical weather predictions" section will be re formulated as follows. The two new tables will be added as supplementary material.*

*MeteoSwiss developed a configuration of the COSMO model (Steppeler et al. 1998) with 1.1 km grid spacing, the COSMO-1 (Fuhrer et al., 2014). It runs as deterministic model and is initialised from its own assimilation cycle using the nudging scheme. Forecasts are calculated*

*every three hours in a rapid update cycle with a forecast range of 33 hours and once per day (03 UTC forecast) out to 45 hours. This setting was operationalised in spring 2016 and replaced the former COSMO-2 with 2.2 km grid spacing and a configuration similar to that described by Baldauf et al., 2011. Configuration changes from COSMO-2 to COSMO-1 are listed in Table S1. As its predecessor, COSMO-1 assimilates radar-derived QPE using latent head nudging. Latent heat nudging is able to considerably increase the accuracy of the precipitation forecast during the first 6 to 12 hours of the forecast. The boundary conditions are taken from the newest available ECMWF (European Centre for Medium-Range Weather Forecasts) high resolution forecast (HRES).*

*Table S1: Configuration of the deterministic models COSMO-2 and COSMO-1*

| *Configuration* | *COSMO-2* | *COSMO-1* |
|---|---|---|
| *Grid spacing* | *2.2 km* | *1.1 km* |
| *Levels* | *60* | *80* |
| *Convection parameterization* | *Shallow convection* | *None* |
| *External parameter fields from* | *GLOBE* | *ASTER* |
| *Num. diffusion for wind* | *On* | *Off* |
| *Boundary conditions* | *COSMO-7* | *IFS HRES* |

*In addition to the deterministic COSMO-1, the ensemble system COSMO-E with 2.2 km grid spacing was operationalised in May 2016 (Klasa et al., 2018). It is initialised twice per day and has a lead time of 120 hours. The assimilation cycle uses an ensemble transform Kalman filter approach (Schraff et al., 2016). The boundary conditions are taken from randomly selected 20 members of the ECMWF ensemble forecast (ENS). It uses the SPPT scheme to simulate the effect of the model uncertainty.  At MeteoSwiss, COSMO-E replaces COSMO-LEPS (Marsigli et al., 2005; Montani et al., 2011), which has a lower resolution with 7 km grid spacing. These and further configuration changes are listed in Table S2.*

Table S2 Configuration of the ensemble prediction models COSMO-LEPS and COSMO-E

| Configuration | COSMO-LEPS | COSMO-E |
|---|---|---|
| Grid spacing | 7.0 km | 2.2 km |
| Levels | 40 | 60 |
| Ensemble scheme | Parameter perturbation (PP) | Stochastically perturbed |

| | | physics tendencies (SPPT) |
|---|---|---|
| Convection parameterization | Tiedkte | Shallow convection only |
| Subgrid scale orography | On | Off |
| Latent head nudging | Off | On (first 2 h) |
| Initial conditions | Interpolated from IFS ENS | Nudging analysis cycle |
| Boundary conditions | IFS ENS | IFS ENS |

**Pag.7, Lines 23-27: Is the confguration of the COSMO-based ensemble changed with the increase o fhorizontal resolution(from 7km of COSMO-LEPS to2.2 kmof COSMO-E)? Authors should add details and references about this issue.**

*AR: see rely to the previous point.*

**Pag.8, Lines 1-3: This sentence is not clear.**

*AR: We will re-arrange it to improve our message.*

**Pag.8, Lines 6-7: Does a threshold exist to identify major food events (namely, food events which are of interest for the authority in charge of the public safety)? How many major food events occurred in summer 2016 for the study area?**

*AR: Such thresholds are published by the SWISS FOEN (https://www.hydrodaten.admin.ch/en/2605.html). In 2016 there was one observed value il danger level 2 (450 m3/s). In 2018 the highest discharge until October 3$^{rd}$ was 157 m3/s, in April). In 2017 the biggest event was 2017 m3/s. All in all a quite flood poor three year span in that region., with 2016 being the most flashy season.*

**Pag.8, Lines 23-24: The comparison may results as not fully proper, given that the observed rainfall input is different for the two chains. Why has the same input not been used for both the chains?**

*AR: The PREVAH-HRU chain is using the same input rainfall from observations in real-time since 2007, and for this reasons we use it as benchmark. RGM-PRO has been designed to work with the most advanced rainfall product of MeteoSwiss (Combiprecip). An assessment on the difference between the two rainfall inputs is presented in Andres et al. (2016, referenced in the manuscript). We will add a sentence to comment on this.*

**Pag.9, Line 13: Is the hourly runoff climatology of the period May-August 2016 statistically meaningful with respect to a longer climatology (for instance, some decades) for the selected study area? It could be useful to show a comparison between the reference climatology of this study and a historical ones for the selected catchment.**

*AR: Good point. We will present this analysis in the revised manuscript .FOEN published such statistics online (https://www.hydrodaten.admin.ch/lhg/sdi/jahrestabellen/2605Q_16.pdf)*

**Pag.9, Lines 14-16: Are the statistical scores computed at each hourly time step of the simulation event? I mean, is the threshold exceedance evaluated each hour and the corresponding score computed at an hourly time step, then averaged over each lead time window? Or is the threshold exceedance evaluated just one time within the whole lead time window? Please clarify.**

*AR: Each hour of each forecast is evaluated by itself and its lead time since start of the forecasts is tracked as attribute. Integral scores are later averaged for all forecasted hours with identical lead time. This explanation will be added to the section.*

**Pag.10, Lines 10-13: The description of the Brier Score decomposition could be omitted, given that it is not discussed in the main manuscript.**

*AR: We decided to keep the definition of the components in the manuscript, as the results are presented in the supplementary material.*

**Pag.11, Lines 20-21: Which is the spatial domain (Switzerland? Verzasca catchment?) over which the scores shown in Fig.3 were computed? Please specify. The scores for POD and FAR are not so satisfying, especially for the higher thresholds (namely, rainfall events which likely trigger food peaks). The FAR scores may results quite high with respect to the usefulness for operational decisions of civil protection authorities. Could authors add a comment to justify this results? Which is the rainfall threshold that trigger major flood events for the study catchments?**

*AR: MeteoSwiss had a very short phase where both systems could be operated and maintained. These are the numbers averaged over Switzerland that have been obtained. In our purpose, this should show the continuity that the new models offer with respect to the predecessors. Floods are triggered by rather intensive rainfall events (above 20 mm per hour). Rainfall based thresholds are currently not used for flash-flood warning in Switzerland.*

**Pag.12, Lines1-2:Why are the BSS values not shown in Fig.4 (or discussed) for the thresholds higher than 10 mm (as done in Fig.3)?**

*AR: The presented rainfall statistic originates from the standard output of the diagnostic and verification routines by MeteoSwiss, where 10 mm/h is the highest threshold delivered for the Brier Skill Score.*

**Pag.12, Lines 14-15: Please specify that the cited scores refer to runoff data and the quantiles refer to the hourly runoff climatology of summer 2016 (in case of my interpretation is right).**

*AR: Will be done.*

**Pag.13, Line 14: The panels "b" and "c" of Fig.8 are very friendly to convey the best performing method, but this visualization does not allow to evaluate if the difference of performance is significant in term of ROCa.**

*AR: You are right. These panels integrates Figure S1 of the supplementary material, where you can better see how much one of the systems outperforms the other one. Figure 8 is specifically designed to answer the question on the best performing method. The link between S1 and Figure 8 will be strengthened and we will elaborate in this respect on the issue of significance.*

**Pag.13, Lines 16-19: Authors should try to justify this result. May the reason for this result be ascribed to the larger dimension of the Verzasca catchment (with respect to its sub-catchment), which can allow to "average" spatial errors in the localization of the rainfall feld?**

*AR: Thanks for the remark. We will elaborate on this.*

**Pag.13, Lines 21-22: The magnitude of the event (i.e., 2-yr return period) should also be cited here (as done at Pag.14, Line 4). Authors could add a comment about the frequency and amount of the flood peaks corresponding to more intense events for the selected catchments.**

*AR: We agree, as already commented. We add here, that is already "good" to have a 2-years event in the selected years, as the two years afterwards no such event occurred.*

**Pag.13, Lines 24-30: The significance of the comparison may result as weak, given that it is discussed for two forecasting chains that differ for the model and input used. Authors should add a comment in order to justify this result with respect to model characteristics and data input.**

*AR: This follows a previous comment on the rainfall input. The reply to the previous point will be extended also here.*

**Pag.15, Line 6: With this statement authors recognize that the discussed results and conclusions may be invalidated by the limited dataset that has been used to test the proposed new forecasting chain. Actually, the usefulness and added value of the new forecasting chain are questionable due to the limited test period. The use of a different dataset could provide different (and opposing) results.**

*AR: We think that "invalidated" not the right term here. As stated before, the chains are different, one of do not rely on calibration, and, during the same period and same constraints, similar skill is found. For us this is an advance with respect to other published approaches for ungauged areas, that have been never benchmarked against state-of-the-art chains. Other authors have been working for years on single extr4me events that have been re-forecasted with and without numerical models, our new approach is quasi-operational.*

*We dared it, we get a promising result, we acknowledge that a short period is a limiting factor, but we think this is a useful communication.*

**Pag.16, Lines 1-19: The detailed comment on the comparison results does not lead to a clear conclusion about which chain should be preferable. Some scores provide opposing outcomes (for instance, in contrast to the companion paper, the process-based forecasts are not better in the sub-catchment), then this analysis may result as inconclusive.**

*AR: See previous comment. We are not looking the best method, best would be to have overall the possibility of using calibrated model. We show that the RGM-PRO approach can compete with a calibrated model in real-time mode. This is for us enough conclusive as compared to the current literature.*

**Pag.19, Line 25:Authors should specify the magnitude of these 22events with respect to the catchment climatology.**

*AR: See previous replies*

**Pag.20, Lines 1-4: This result is quite general and recalls several past studies. Here, it appears as a local application based on a limited dataset.**

*AR: We will further hint at the limitations emerged.*

**Pag.20, Lines 5-12:Authors recognize the major drawback of the present study. They would evaluate a new model-based approach to flood forecasting which does not re-quire the calibration task for the hydrological module, but the available dataset is not appropriate (due to the very limited size) to prove the added value of the proposed approach.**

*AR: Again, we are not seeking for added value, but for a useful tool that do not require calibration. The results with a limited set of data show us that we are on the right way.*

References:

Addor, N., Jaun, S., Fundel, F., and Zappa, M.: An operational hydrological ensemble prediction system for the city of Zurich (Switzerland): Skill, case studies and scenarios, Hydrology and Earth System Sciences, 15, 2327–2347, doi:10.5194/hess-15-2327-2011, 2011.

Antonetti, M., Horat, C., Sideris, I. V., and Zappa, M.: Ensemble flood forecasting considering dominant runoff processes: I. Setup and application to nested basins (Emme, Switzerland), Nat. Hazards Earth Syst. Sci. Discuss., https://doi.org/10.5194/nhess-2018-118, in review, 2018.

Cane, D., Ghigo, S., Rabuffetti, D., and Milelli, M.: Real-time flood forecasting coupling different postprocessing techniques of precipitation forecast ensembles with a distributed hydrological model. The case study of may 2008 flood in western Piemonte, Italy, Nat. Hazards Earth Syst. Sci., 13, 211-220, https://doi.org/10.5194/nhess-13-211-2013, 2013.

Devoli, G., Tiranti, D., Cremonini, R., Sund, M., and Boje, S.: Comparison of landslide forecasting services in Piedmont (Italy) and Norway, illustrated by events in late spring 2013, Nat. Hazards Earth Syst. Sci., 18, 1351-1372, https://doi.org/10.5194/nhess-18-1351-2018, 2018.

Kobayashi, K., Otsuka, S., Apip, and Saito, K.: Ensemble flood simulation for a small dam catchment in Japan using 10 and 2 km resolution nonhydrostatic model rainfalls, Nat. Hazards Earth Syst. Sci., 16, 1821-1839, https://doi.org/10.5194/nhess-16-1821-2016, 2016.

Li, Z., Li, Y., Bonsal, B., Manson, A. H., and Scaff, L.: Combined impacts of ENSO and MJO on the 2015 growing season drought on the Canadian Prairies, Hydrol. Earth Syst. Sci., 22, 5057-5067, https://doi.org/10.5194/hess-22-5057-2018, 2018.

Picciotti, E., Marzano, F. S., Anagnostou, E. N., Kalogiros, J., Fessas, Y., Volpi, A., Cazac, V., Pace, R., Cinque, G., Bernardini, L., De Sanctis, K., Di Fabio, S., Montopoli, M., Anagnostou, M. N., Telleschi, A., Dimitriou, E., and Stella, J.: Coupling X-band dual-polarized mini-radars and hydro-meteorological forecast models: the HYDRORAD project, Nat. Hazards Earth Syst. Sci., 13, 1229-1241, https://doi.org/10.5194/nhess-13-1229-2013, 2013.

Philipp, A., Kerl, F., Büttner, U., Metzkes, C., Singer, T., Wagner, M., and Schütze, N.: Small-scale (flash) flood early warning in the light of operational requirements: opportunities and limits with regard to user demands, driving data, and hydrologic modeling techniques, Proc. IAHS, 373, 201-208, https://doi.org/10.5194/piahs-373-201-2016, 2016.

Zappa, M., Jaun, S., Germann, U., Walser, A., and Fundel, F.: Superposition of three sources of uncertainties in operational flood forecasting chains, Atmospheric Research, 100, 246–262, doi:doi:10.1016/j.atmosres.2010.12.005, 2011.

---

## Author Comment (AC2) · 11 Oct 2018

**Ensemble flood forecasting considering dominant runoff processes: II. Benchmark against a state-of-the-art model-chain (Verzasca, Switzerland)"**

Authors replies to RC1 (Eric Gaume):

We want to thank Eric Gaume for his assessment of our manuscript. In the following we give our answers to the comments and recommendations that have been raised. Reviewer comments RC are **bold,** our reply AR is in *italic*. Insertions in the revised manuscript MI are underlined. This reply is in some parts identical with the already uploaded reply to reviewers 2, as some issues have been raised by both reviewers.
* * *
**GENERAL ASSESSMENT**

**RC: The connection with real-world operational methods and application is a very positive aspect of the presented work. To my knowledge, particularly advanced methods are implemented operationally in Switzerland if compared to the rest of the world. Nevertheless, the proposed comparison is conducted for one single watershed (and one of its sub-watersheds but the results are not presented in the manuscript) and for a period of time of 3 months only (May to August 2016). This is by far insufficient to draw real convincing conclusions…… Likewise, a three months period seems far too short for a faithful evaluation of forecasting models and does not correspond to the general standards of the scientific publications. The results may be too dependent on some few if not a single flood event with no possibility of generalization. The authors themselves acknowledge in the discussion part of their manuscript that sparse data may be problematic (P 18 L6)…. The objectives and methods presented in the paper are correct, but the manuscript can hardly be published to my opinion unless a much larger data set in time and space (larger period of time and larger number of watershed is considered). The team seems to have access to reach data sets in Switzerland ; It is time consuming of course, but I do not see any reason why they could not conduct a large and necessary test and validation study based on the approach presented in the manuscript.**

*AR: This issue has been raised also by the reviewer 2 and by the reviewers of the companion paper by Antonetti et al. (2018). We are of course aware, that more basins and longer periods of evaluation are always welcome. NHESS (but also HESS) is in this respect a journal that regularly publishes case studies (e.g. Kobayashi et al., 2016; Cane et al., 2013), preliminary assessments (Picciotti et al., 2013) or intercomparison of approaches during limited period of time (e.g. Davoli et al., 2018; Li et al., 2018). Having targeted NHESS as journal for disseminating our experience, this study is designed to benchmark a state-of-the-art and operational calibrated hydrological prediction system(PREVAH-HRU) against a newly developed event-based system that is configured without calibration requirements (RGM-PRO). This has been evaluated during a representative flood season and in case a nested*

*basins where PREVAH-HRU simulations have been running and collected in real-time (so no tailored experiment here, 100% data from a deployed system), while RGM-PRO simulations have been completed as reforecast experiment in the framework of a master project (August 2016 to February 2017) by the lead author. With this approach we can learn about the quality of the novel approaches from different perspectives at the same time. The transfer of experience to another catchment and climatic region is presented in the companion paper by Antonetti et al. (2018), while in prior studies we shown how PREVAH-HRU behave in case of long-term analyses (e.g. Addor et al., 2011 and Zappa et al., 2011). This is also why we take so much time and space in order to discuss these findings with respect to our previous studies. Furthermore, as far as the length of the investigation period is concerned, some limitations arise from the use of COSMO-E and COSMO-1. MeteoSwiss decommissioned after several years the antecedent operational NWP COSMO-2 and COSMO-LEPS in 2016. As we want to make our systems operational, it was for us important to focus on a first analysis with the new NWPS that we receive and archive in real-time since February 2016. Even Switzerland cannot afford to generate re-forecasts of ensemble weather predictions, and only a limited time was available to compare the old atmospheric EPS system (COSMO-LEPS) with the new one (COSMO-E). As COSMOE is the future, we decided to focus on the available COSMOE data.*

*In the revised manuscript we will try to indicate to which extent the available season is representative with respect to log-term discharge observations in the target area. We have currently unfortunately no capacity to extend the analyses beyond 2016, and in case of both 2017 and 2018, this would not really beneficial since no severe flood event occurred (see also our detailed reply to reviewer 2). Summarizing, for the time being we are not able to generate "useful" additional runs of our event-based model RGM-PRO.*

*Here we are not looking for added value with respect to the traditional forecasts, but for a useful tool that do not require calibration. The results with a limited set of data show us that we are on the right way.*

*For this study, we selected two different chains are, one of do not rely on calibration, and, during the same period and same constraints, similar skill is found. For us this is an advance with respect to other published approaches for ungauged areas, that have been never benchmarked against state-of-the-art chains. Other authors have been working for years on single extreme events that have been re-forecasted with and without numerical models, our new approach is quasi-operational. We dared it, we get a promising result, we acknowledge that a short period is a limiting factor, but we think this is a useful communication.*

**RC: The implementation of a rainfall-runoff model without calibration can only be evaluated if conducted on a significant number of watersheds – typically some tens.**

*AC: RGM-PRO has been introduced by Antonetti et al. (2017) with an analysis of 5 basins and 8 events. In Antonetti et al. (2018) three additional basins have been implemented and discussed. In this manuscript RGM-PRO is configured (not calibrated) for the Verzasca to*

increase (by two, with the nested basin) the number of applications. The Verzasca basin is one of four basins where we could have performed such a benchmark with "traditional" operational forecasts, the other three being sub-areas of the Sihl river (Addor et al., 2011), for which have by far less published work on flash-flood than in case of the Verzasca.

This table has been also provided to the reviewers of our companion paper to show how these papers relate to our previous studies. The table will be included in the companion paper by Antonetti et a.l (NHESSD).

| Paper | Zappa et al. | Addor et al. | Liechti et al. | Antonetti et al. | Antonetti et al. | Antonetti et al. | Horat et al. |
|---|---|---|---|---|---|---|---|
| Year | 2011 | 2011 | 2013 | 2017 | 2018 | 2018 | 2018 |
| Journal | At. Research | HESS | HESS | Hydrol. Proc. | HESS | NHESSD | NHESSD |
| **Target areas** | | | | | | | |
| Verzasca | X | | X | | | | X |
| Sihl | | X | | | | | |
| Emme | | | | | X | X | |
| Other | | | X | X | | | |
| **Topics** | | | | | | | |
| Forecasting | X | X | X | | | X | X |
| Model development | | | | X | | | |
| Uncertainty propagation | X | | X | | X | (X) | (X) |
| Intercomparison | | X | X | (X) | (X) | X | X |
| **Model/module** | | | | | | | |
| PREVAH-HRU | X | X | X | | | | X |
| RGM-PRO | | | | X | X | X | X |
| RGM-TRD | | | | | X | X | |
| **Rainfall forcing** | | | | | | | |
| Intrepolated gauges | X | X | X | | X | | X |
| Combiprecip | | | | X | X | X | X |
| COSMO-1 | | | | | | X | X |
| COSMO-2 | X | X | X | | | | (X) |
| COSMO-LEPS | X | X | | | | | (X) |
| COSMO-E | | | | | | X | X |
| Weather radar nowcasting | X | | X | | | | |
| Frequency | continuous | continuous | events | events | events | events | events |
| Period | 2007-2010 | 2007-2009 | 2007-2010 | 2005-2016 | 2005-2016 | 2016 | 2016 |
| **Analyses** | | | | | | | |
| NSE/KGE | NSE | | | KGE | KGE | NSE/KGE | |
| Brier/ROC/FAR/RankHist | (X) | X | X | | | X | X |
| MonteCarlo | X | | (X) | X | X | X | (X) |
| Other | SWAE | | | | ANOVA | | |

Zappa et al. (2011) is our benchmark paper on uncertainty propagation

Addor et al.. (2011) is our reference work on verification of deterministic and ensemble forecasts (with the Breir score as main metric to discriminate between deterministic and probabilistic forecasts).

Liechti et al. (2013) focuses on flash-flood nowcasting with advanced weather radar products

Antonetti et al. (2017) introduces RGM-PRO

Antonetti et al. (2018, HESS) evaluate structures and configurations of RGM-PRO in the Emme catchment

Antonetti et al. (2018, NHESSD) first apply RGM-PRO in forecasting mode for the Emme catchment and is our first study with COSMO-E/COSMO-1

Horat et al. (2018, NHESSD) applies RGM-PRO in forecasting mode for the Verzasca catchment and compare its quality with our current operational model as forced by COSMO-E/COSMO-1.

**RC: 1) The authors mostly refer to their own works. Indeed, interesting and innovative methods are implemented in Switzerland to forecast flash floods. But it would important also to cite works conducted in other countries and by other teams on the same issue at least in the introduction of the manuscript to show the originality of the proposed approach. Flash flood forecasting has been an active field of research in the recent years.**

*AR: The reviewer is right. The introduction of this "Part II" was the most difficult section to write, as we already review recent work on flash flood in "Part I". We did not want to replicate great parts of the introduction from "Part I" here and thus focussed on introducing the state-of-the-art of our applications in the Verzasca river. We will try to re-formulate the introduction and include a better-balance between own work, previous work and the review presented for "Part I".*

**RC: 2) The manuscript refers in many places to a companion paper and to supplementary materials. This is frustrating for the readers since some important information is not provided in the manuscript such as the implementation of the "process based" model (what are the input variables, how are the values of the parameters of the model fixed) or the results obtained for the Pincascia sub-watershed. Supplementary material is interesting but a manuscript must be to a certain extent self-sufficient and contain at least the basic information needed for the interpretations and the results that are commented and interpreted.**

*AR: We are of the opinion that "Part I" and "Part II" are together "self-sufficient" but agree that for a reviewer providing the review of "Part II" only, some frustration might arise, because some of the methods have been detailed in "Part I" and previous publications. The document the reviewer assessed is not intended to introduce RGM-PRO (see Table on the previous page). We compare here two published models, one of them do not require calibration and uses the parameterization presented in Antonetti et al. (2017). The PREVAH-HRU chain is (beside the COSMOE/COSMO1 forcing) identical with the chain first presented in Zappa et al., (2011). In the revised version we will try to make this manuscript more self-sufficient without replicating what shown in other papers. As far as the use of supplementary material is concerned, we selected to follow the journals general wish of manuscripts with a reduced number of figures and moved figures showing the same for another area to the supplementary material, while keeping both basins in the synthesis presented in Figure 8. In the revised manuscript we will include the results of Figure 6 and 7 for the Pincascia basin in the main manuscript.*

**RC: 3) Brier scores are used to compare deterministic and probabilistic forecasts. I know that some other papers did the same, but this comparison is not appropriate. Indeed, a Brier score can be computed in both cases, but do not measure exactly the same things and can therefore not be directly compared. Forecasts must be combined with a utility function and evaluated in a decision making context for a proper and rigorous comparison.**

*AR: The Brier Score (BS) is our link to previous studies where we are well confident that BS is a very efficient way to discriminate between the skill of deterministic and probabilistic forecasts. We will elaborate on this statement in the revised manuscript.*
* * *
**COMMENTED DOCUMENT**

**RC: Usually, no refrences are cited in the abstract of a paper**

*AR: We will remove the citation to the companion paper from the abstract.*

**RC: This seems to be a too short period for a real evaluation of forecasting model even if a 2-year flood has been observed during this Summer.**

*AR: See replies to the general comments*

**RC: Such a conclusion cannot be drawn based on two application examples only. Wider applications of the proposed methods would be needed to confirm or invalidate this impression.**

*AR: See replies to the general comments*

**RC: Again, I have here some doubts and in any case no general conclusion ca be drawn from the presented limited test case. Several previous studies, including the so-called DMIP experiment (Distributed model intercomparison program) did not lead to the conclusion that "processed based" models do have better performances than calibrated conceptual models. On the contrary...**

*AR: We are not aiming at declaring that RGM-PRO without calibration is better that the (process-based, calibrated) PREVAH-HRU model with very conceptual runoff generation module. We are happy to see that they show similar quality in a period including a flood with 2-years return period. Thanks for mentioning DMIP. We will include this in the introduction and discussion.*

**RC: Why such a detailled emphasis on Doswell's work ?**

*AR: We will condense this part of the introduction and add work of other authors.*

**RC: Forecasted : they are some spelling errors in the manuscript that need to be corrected.**

*AR: We corrected this occurrence and will look for other typos as suggested by the reviewer.*

**RC: The literature review is mostly citing works conducted in Switzerland by the authors' team. It could be enlarged significantly.**

*AR: See replies to the general comments*

**RC: I am not found of the term process or physically based since the implementation scale of the models is generally much too coarse to enable a detailed description of the hydrological processes. More-over, even the process based model generally need some calibration.**

*AR: RGM_PRO is a dominant runoff process based module for the process of runoff generation. Parameters can be set a priori basing on calibration of sprinkling experiments under different conditions as introduced in detail by Antonetti et al. (2017) and as illustrated in "Part I" of this manuscript. We will expand this section in order to give a better orientation to the readers that are not familiar with our previous study.*

**RC: Is an hourly time step really suited for such a small watershed ?**

*CombiPrecip would be available in 10 minutes step. The disaggregation of COSMO-forecasts into 10 minutes field goes beyond the scope of this paper. According to our experience and for basins in the order of size of the Verzasca are one hour forcing data sufficient. The model internal integration time step is of 10 minutes.*

**RC: Please develop the acronym (FOEN)**

*AR: Already defined on Page 3.*

**RC: Develop the acronym (SPPT)**

*AR: Section 3.2.2 will be expanded as suggested by reviewer 2. All acronyms will be defined.*

**RC: 1 alert every 5 days ! The number of detections and False alarms could be indicated here.**

*AR: Will be done.*

**RC: I am not found of the reference to supplementary material. An article should be self-sufficient. If the information is important for the analysis, it should be provided in the paper.**

*AR: We thank the reviewer for this amendment. A figure will be added and text will be included to close this gap of information.*

[Figure]

*New Figure: Flood hydrographs of reference simulation by RGM-PRO as forced by CombiPrecip (cpc) for three events in the analysis period. Top panels: Verzasca basin. Bottom panels: Pincascia river.*

**RC: The references to the companion paper are too frequent. Again, the manuscript should be self-sufficient and contain the necessary information and results.**

*AR: This sentence can actually be removed and will be removed.*

**RC: The definition of BS could be recalled. I am wondering if BS values computed for deterministic models with binary results (1 exceedance and 0 non-exceedance) and for probabilistic models (probability of exceedance) are comparable ? Binary models may provide systematically larger BS values...**

*AR: … and thus less skill for users needing taking action basing on a threshold. The question is if an ensemble model with coarser resolution is more useful than a high resolution deterministic mode. We will surely expand on the BS issue to clarify our reason for using it.*

**RC: the signification of symbol o should be explained. Add a legend to this equation.**

*AR: Will be done.*

**RC: 15:12:16 BS and BSS can be computed for both types of forecasts. I wonder if the obtained values can directly be compared. Probably not).**

**RC: Again : include important results in the manuscript.**

*AR: This one will stay in the supplementary material.*

**RC: The bootstrappping procedure must be better explained. A few sentences can be added. The reader cannot understand what exactly is resampled.**

*AR: Events are removed from the available sample and the score are recalculated. This allows for quantification of uncertainty stemming from the choice of events. Clarification will be included in the revised manuscript.*

**RC: It is important to mention somewhere in the manuscript how many events are included in each analysed sample.**

*AR: The number of cases per each rainfall intensity evaluated will be provided.*

**RC: Sign of larger discharge values produced by the process-based model. The balance between POD and FAR must be better justified. What are the operational needs. How would end-users judge both results?**

*AR: The traditional model is quite lumped with respect to runoff generation description and thus tends to smooth intense local precipitation in terms of output simulated discharge. The process based RGM-PRO accounts for local impervious areas with rapid transformation of rainfall into discharge and thus yields generally more "flashy" hydrographs. The fact of having high POD is surely an advantage, but high FAR is also not nice from an user perspective. In the end each user should decide his risk profile under consideration of costs of action and remaining risk that he can accept. For this case we have no user to ask, but in case of the Sihl forecast presented in Addor et al. (2011) the user is surely interested of having no missed events and is ready to take action even if an event would not occur in the end. We will elaborate these thoughts in the manuscript.*

**RC: Some explanation of the contrasted behaviour of both models could be interesting for the reader.**

*AR: We will elaborate this remark.*

**RC: Or even much more, since results for larger lead times are not shown. The stability of the skill score with lead times is a surprising result a deserves some comments.**

*AR: COSMO1 lead time do not go beyond the shown range. COSMOE does and we show the results for the lead times beyond hour 29 in Figure 8.*

**RC: This is what is generally expected**

*AR: Yes, whereby in some cases ensembles need some hours to develop and best skill is obtained between 24 and 48 hours lead time.*

**RC: That is less systematic**

*AR: Yes.*

**RC: This very surprising result should not only be described but analysed...**

*AR: We think that this is due to the fact, that Addor et al. (2011) do not evaluate at sub-daily scale and thus the "constant" skill in the first 36 hours is not intelligible. Liechti et al. (2013) do not use ensemble EPS, and thus the increase of skill in the first days in not shown for an ensemble system. Such behaviour has been described for the Verzasca in Zappa et al. (2013, Figure 3.), where maximum skill (ROCa) was between day 3 and 4.*

**RC: Yes, the presented results may be extremely dependent on the specificities of the analyzed sample that is of limited size. Is it really possible to draw valuable general conclusions ?**

*AR: Yes, it is, because as already stated both chain "suffers" from a limited sample size.*

**RC: Again a strange results. How many events control the criteria values for large thresholds ?**

*AR: Very few.*

**RC: This may depend on the antecedent moisture conditions**

*AR: The reviewer might be right, but as already discussed, we think that the lumped structure in runoff generation description of PREVAH-HRU leads to a general "inertia" during the wetting phase.*

**RC: Unclear sentence**

*AR: Re-formulation: "Furthermore, the process-based forecasting chains react with less delay to rainfall input, leading to higher peaks in runoff but also larger uncertainties when applied for ensemble forecasting. Although the use of information about DRP decreases the hydrological model parameter uncertainty, as found by Antonetti et al. (2016b), it does not decrease the total uncertainty when ensemble precipitation forecasts are used. In other words the fully distributed DRP-approach amplifies the spread originating from different members of a precipitation ensemble, while the semi-distributed PREVAH-HRU approach strongly smooths such differences.*

**RC: The authors are comparing their work to their previous works... Is this interesting for a broader audience ?**

*AR: If the reader is interested on performance of an uncalibrated model as compared to the one of a calibrated model then yes. We will re-arrange sections 6.3 to 6.6 as suggested by reviewer 2.*

**RC: Please provide some explanation to this observed result**

*AR: "Sample size", but maybe also skill of the DRP approach in case of larger flood peaks.*

**RC: Readers that are not familiar with the author's work can hardly follow. What is the aim of comparing the two published results if according to the authors the methods are too different to really enable a comparison ?**

*AR: Our original formulation might suggest that we are looking for the "best" approach, while as stated before, we want to show that comparable results can be achieved without calibration. We will adapt this sentence in order to be compliant with our goals.*

**RC: Again, I am not sure that both BSS values can be compared. Discrepancies between forecasts and observations lead to BS contributions equal to 1 for discrete models, while contributions are always much lower than one (difference at a power two) for ensemble approaches.**

*AR: We see the point of the reviewer and can suggest to this issue this blog on HEPEX: https://hepex.irstea.fr/how-can-the-brier-score-know-my-inner-thoughts/ Even if we will not make a "Brier" paper out of it we will add more thoughts on its use for comparing deterministic and ensemble forecasts.*

**RC:, that is a major issue, and the data set used in this study is extremely sparse (3 months). Are the obtained results really meaningful and worth an interpretation. I really have major doubts.**

*AR: This re-iterates previous comments on this issue.*

**RC: Yes, that is also very true and would provide additional elements for the interpretation of the results. The authors have the data, why did they not conduct this analysis and include it in the manuscript ?**

*AR: As also replied to reviewer 2, the output diagnostic from COSMOE and COSMOE is automated and re-configuration would have been beyond the scope of the paper. In the meantime a paper on the verification of COSMOE has been presented by Klasa et al. (2018)*

**RC: Is this a really significant result or does it illustrate the consequence of the dependence of the result to the limited sample (sampling variability) ?**

*AR: Significant at a very preliminary level.*

**AR: This is a pure speculation at this stage. This hypothesis needs to be confirmed on a larger set of test cases before it can really be formulated.**

*AR: We will formulate the sentence to render its speculative character.*

**AR: No model can be really described as process-based or physically-based. Ideally the term should be in quotation marks...**

*AR: Will be done.*

**RC: Perhaps "with no calibration" or "implemented without being calibrated", but to my experience and the experience of most hydrological modelers, every model benefits from calibration...**

*AR: We agree that each model can benefit from calibration, when observations are available. RGM-Pro process-based reaction to precipitation has been calibrated against sprinkling experiments (Antonetti et al., 2017). This relation is now used as "universal" parameterization for any application of RGM-PRO. The mapping of the areas where*

*processes occur determine how specific catchments reacts. We mapped the Verzasca and applied the RGM-PRO parametrization without any further calibration.*

**RC: intensely**

*AR: Thanks for the correction.*

**RC: Yes, this is my major critic. I read this sentence as a clear understatement. Is not fully appropriate or fully inappropriate in the sense that the data set is far insufficient to draw clear conclusions.**

*AR: See our previous replies.*

**RC: Some elements are missing explaining the implementation principles of the new approach. The new model has also parameters. What source of data has been used to fix the value of these parameters ?**

*AR: See previous replies (sprinkling experiments) and Antonetti et al. (2017).*

References:

Addor, N., Jaun, S., Fundel, F., and Zappa, M.: An operational hydrological ensemble prediction system for the city of Zurich (Switzerland): Skill, case studies and scenarios, Hydrology and Earth System Sciences, 15, 2327–2347, doi:10.5194/hess-15-2327-2011, 2011.

Antonetti, M., Horat, C., Sideris, I. V., and Zappa, M.: Ensemble flood forecasting considering dominant runoff processes: I. Setup and application to nested basins (Emme, Switzerland), Nat. Hazards Earth Syst. Sci. Discuss., https://doi.org/10.5194/nhess-2018-118, in review, 2018.

Cane, D., Ghigo, S., Rabuffetti, D., and Milelli, M.: Real-time flood forecasting coupling different postprocessing techniques of precipitation forecast ensembles with a distributed hydrological model. The case study of may 2008 flood in western Piemonte, Italy, Nat. Hazards Earth Syst. Sci., 13, 211-220, https://doi.org/10.5194/nhess-13-211-2013, 2013.

Devoli, G., Tiranti, D., Cremonini, R., Sund, M., and Boje, S.: Comparison of landslide forecasting services in Piedmont (Italy) and Norway, illustrated by events in late spring 2013, Nat. Hazards Earth Syst. Sci., 18, 1351-1372, https://doi.org/10.5194/nhess-18-1351-2018, 2018.

Klasa C, Arpagaus M, Walser A, Wernli H. An evaluation of the convection-permitting ensemble COSMO-E for three contrasting precipitation events in Switzerland. Q J R Meteorol Soc. 2018;144:744–764. https://doi.org/10.1002/qj.3245

Kobayashi, K., Otsuka, S., Apip, and Saito, K.: Ensemble flood simulation for a small dam catchment in Japan using 10 and 2 km resolution nonhydrostatic model rainfalls, Nat. Hazards Earth Syst. Sci., 16, 1821-1839, https://doi.org/10.5194/nhess-16-1821-2016, 2016.

Li, Z., Li, Y., Bonsal, B., Manson, A. H., and Scaff, L.: Combined impacts of ENSO and MJO on the 2015 growing season drought on the Canadian Prairies, Hydrol. Earth Syst. Sci., 22, 5057-5067, https://doi.org/10.5194/hess-22-5057-2018, 2018.

Picciotti, E., Marzano, F. S., Anagnostou, E. N., Kalogiros, J., Fessas, Y., Volpi, A., Cazac, V., Pace, R., Cinque, G., Bernardini, L., De Sanctis, K., Di Fabio, S., Montopoli, M., Anagnostou, M. N., Telleschi, A., Dimitriou, E., and Stella, J.: Coupling X-band dual-polarized mini-radars and hydro-meteorological forecast models: the HYDRORAD project, Nat. Hazards Earth Syst. Sci., 13, 1229-1241, https://doi.org/10.5194/nhess-13-1229-2013, 2013.

Zappa, M., Jaun, S., Germann, U., Walser, A., and Fundel, F.: Superposition of three sources of uncertainties in operational flood forecasting chains, Atmospheric Research, 100, 246–262, doi:doi:10.1016/j.atmosres.2010.12.005, 2011.